# CRISPR-based functional genomics in human dendritic cells

Marco Jost[1,2,3,4†‡*], Amy N Jacobson[5,6†], Jeffrey A Hussmann[1,2,3,4,7], Giana Cirolia[8], Michael A Fischbach[5,6,8*], Jonathan S Weissman[1,2,3,7,9*]

[1]Department of Cellular and Molecular Pharmacology, University of California, San Francisco, San Francisco, United States; [2]Howard Hughes Medical Institute, University of California, San Francisco, San Francisco, United States; [3]California Institute for Quantitative Biosciences, University of California, San Francisco, San Francisco, United States; [4]Department of Microbiology and Immunology, University of California, San Francisco, San Francisco, United States; [5]Department of Bioengineering, Stanford University, Stanford, United States; [6]ChEM-H, Stanford University, Stanford, United States; [7]Whitehead Institute for Biomedical Research, Cambridge, United States; [8]Chan Zuckerberg Biohub, San Francisco, United States; [9]Department of Biology, Massachusetts Institute of Technology, Cambridge, United States

*For correspondence:
marco_jost@hms.harvard.edu (MJ);
fischbach@fischbachgroup.org (MAF);
weissman@wi.mit.edu (JSW)

†These authors contributed equally to this work

Present address: ‡Department of Microbiology, Harvard Medical School, Boston, United States

**Abstract** Dendritic cells (DCs) regulate processes ranging from antitumor and antiviral immunity to host-microbe communication at mucosal surfaces. It remains difficult, however, to genetically manipulate human DCs, limiting our ability to probe how DCs elicit specific immune responses. Here, we develop a CRISPR-Cas9 genome editing method for human monocyte-derived DCs (moDCs) that mediates knockouts with a median efficiency of >94% across >300 genes. Using this method, we perform genetic screens in moDCs, identifying mechanisms by which DCs tune responses to lipopolysaccharides from the human microbiome. In addition, we reveal donor-specific responses to lipopolysaccharides, underscoring the importance of assessing immune phenotypes in donor-derived cells, and identify candidate genes that control this specificity, highlighting the potential of our method to pinpoint determinants of inter-individual variation in immunity. Our work sets the stage for a systematic dissection of the immune signaling at the host-microbiome interface and for targeted engineering of DCs for neoantigen vaccination.

## Introduction

Dendritic cells (DCs) play an outsized role in orchestrating innate and adaptive immunity: they act as sentinels, detecting invaders and initiating innate immune responses to clear them, and as antigen-presenting cells, initiating adaptive immune responses that are antigen-specific and tailored to the context in which the antigen was detected (*Merad et al., 2013*; *Sun et al., 2020*). In this fashion, DCs mediate pathogen clearance, tumor cell killing, and tolerance to microbiome bacteria or dietary antigens. DCs thus play fundamental roles in shaping host-pathogen and host-microbiome interactions and in the etiology of autoimmune disorders and are a major target for efforts to develop new generations of immunotherapies (*Wculek et al., 2020*).

Dissecting the pathways by which human DCs respond to innate immune stimuli and relay them into adaptive responses, however, has been challenging, due in large part to difficulties in genetically manipulating human DCs. Although approaches for gene repression in human DCs by RNAi have been reported (*Song, 2014*), RNAi suffers from limited efficacy and specificity, precluding broader implementation (*Kaelin, 2012*). As a consequence, DC biology is generally studied in mouse

models, but mice and humans differ in many aspects of both innate and adaptive immunity, including innate immune receptor repertoires, responses to immune ligands such as lipopolysaccharide (LPS), and developmental pathways of adaptive immune cells (*Pulendran and Davis, 2020*). One way to address this challenge is to knock out genes in DC precursor populations such as monocytes or stem cells, followed by differentiation into DCs (*Freund et al., 2020*; *Hiatt et al., 2020*; *Laustsen et al., 2018*). These methods, however, require independent differentiation of each knockout population and as a result are susceptible to batch effects and poorly suited for genetic screens. Moreover, they do not permit probing the functions of genes required for DC differentiation and culture. More broadly, both animal and stem cell models fail to capture inter-individual variation in immune phenotypes (*Sanz et al., 2018*), which has been observed for example in innate immune responses, autoimmunity, and pathogen susceptibility (*Brodin and Davis, 2017*; *Fairfax et al., 2014*; *Ye et al., 2014*) and has gained further salience during the Covid-19 pandemic (*Lucas et al., 2020*; *Pereira et al., 2021*). Such variation results from a combination of genetic factors and lifelong environmental exposures (e.g., from the microbiome) but it remains challenging to define the causal determinants in the absence of genetic tools for patient-derived immune cells such as DCs.

To address these limitations, we developed a CRISPR-Cas9 strategy to construct targeted knockouts directly in human monocyte-derived DCs (moDCs), which are readily derived from donor blood and are widely used for research and clinical applications (*Sallusto and Lanzavecchia, 1994*; *Garg et al., 2017*). Using this strategy, we conducted a genetic screen for factors that recognize an innate immune ligand from the human microbiome, LPS from the gut commensal *Bacteroides thetaiotaomicron* (*B. theta*), recapitulating known LPS signaling pathways and revealing candidate genes that mediate species-specific LPS recognition and give rise to inter-individual variation in the response to LPS. These results highlight the potential of our strategy for identifying receptors for innate immune ligands, such as those from the human microbiome, and for pinpointing the genetic bases of inter-individual variation in human immunity. More broadly, our work now provides a general blueprint for functional genomics in human DCs.

## Results

### A CRISPR-Cas9 strategy for functional genomics in human moDCs

To enable introduction of specific knockouts in human moDCs, we developed a non-viral genome editing strategy based on electroporation of in vitro-assembled Cas9-sgRNA complexes (Cas9 ribonucleoprotein particles, RNPs), an approach that has been validated in other immune cell types (*Freund et al., 2020*; *Hiatt et al., 2020*; *Riggan et al., 2020*; *Roth et al., 2018*; *Schumann et al., 2015*). Briefly, our strategy entails isolating monocytes from human donor blood, differentiating them into moDCs in the presence of GM-CSF and IL-4, and electroporating these moDCs with Cas9 RNPs to induce double-strand breaks at the targeted locus (*Figure 1a*). Such double-strand breaks trigger error-prone DNA repair and subsequent formation of insertions or deletions (indels) that, with a certain frequency, cause frameshift mutations and thus knockout of the targeted gene. We monitor the efficiency of this process by genotyping using next-generation sequencing and by phenotyping using functional assays.

We first electroporated moDCs with Cas9 RNPs targeting *AAVS1*, using a validated sgRNA sequence (*Mali et al., 2013*), as well as *TNF* and *TLR4* with sgRNA sequences from the Brunello library (*Supplementary file 1*; *Doench et al., 2016*). By testing a grid of electroporation conditions, we identified conditions with efficient genome editing of *AAVS1* and *TNF* and limited toxicity, but editing of *TLR4* was inefficient (*Figure 1b,c*, *Figure 1—figure supplement 1a,b*, *Supplementary file 2*). The large majority of observed indels were 1 or 2 bp deletions (*Figure 1d*, *Figure 1—figure supplement 1c*), which are frameshift mutations that eliminate the function of the gene.

To improve editing efficiency, we (i) leveraged sgRNA design tools optimized for RNP activity (CRISPR Design Tool, Synthego) and (ii) targeted each locus with two to three sgRNAs with binding sites tiled across a 200 bp stretch to induce simultaneous double-strand breaks, a design that increases the likelihood of achieving functional knockouts by preventing error-free DNA repair and/or removing a stretch of the gene (Materials and methods) (*Riggan et al., 2020*). Because such large deletions generate smaller amplicons in our genotyping approach, which may be overrepresented in

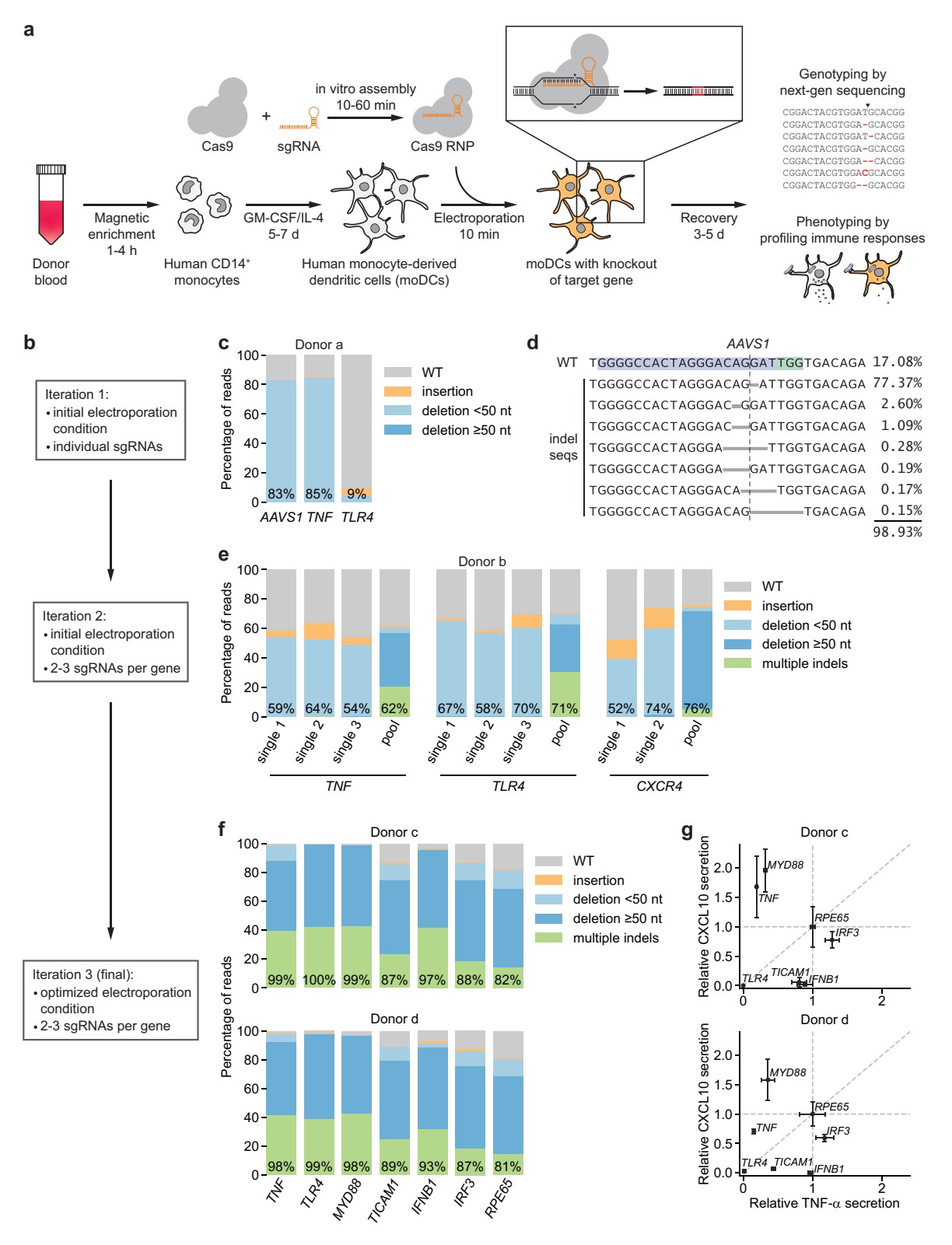

**Figure 1.** CRISPR/Cas9-based strategy for gene knockout in human moDCs. (**a**) Schematic outline of the strategy. (**b**) Flow chart delineating optimization. (**c**) Percentage of genomic DNA reads assigned to different classes of outcomes after targeting each locus with Cas9 RNPs in condition P1, CB-150. Labeled values indicate total percentage of reads with a non-WT sequence. (**d**) Individual editing outcomes at the *AAVS1* locus accounting for at least 0.15% of on-target reads. Horizontal gray bars denote deletions. Vertical dashed line denotes Cas9 cut site. (**e**) Outcome classification, as in

*Figure 1 continued on next page*

*Figure 1 continued*

(c), after targeting indicated loci with single or multi-sgRNA Cas9 RNPs. (f) Outcome classification, as in (c), after targeting seven loci with multi-sgRNA Cas9 RNPs in moDCs from two independent donors. (g) Production of TNF-α and CXCL10 by knockout moDCs challenged with 100 ng/mL *E. coli* O55 LPS, normalized to cell numbers and to cytokine production in moDCs with knockout of *RPE65*. Data represent mean and standard deviation of two independent treatments for both TNF-α and CXCL10 levels. See also *Figure 1—figure supplement 1*, *Figure 1—figure supplement 2*, *Figure 1—figure supplement 3*, *Figure 1—figure supplement 4*, and *Figure 1—figure supplement 5*.

The online version of this article includes the following figure supplement(s) for figure 1:

**Figure supplement 1.** Additional characterization of initial conditions for moDC genome editing strategy.
**Figure supplement 2.** Strategy to measure effect of amplicon size on observation (amplification + sequencing) efficiency.
**Figure supplement 3.** Identification of an optimal electroporation condition for moDC genome editing.
**Figure supplement 4.** Schematic overview of TLR4 signaling pathways.
**Figure supplement 5.** Immunophenotyping of edited moDCs.

sequencing counts due to length biases in PCR amplification and the sequencing reaction itself, we devised a scheme to correct sequencing counts for length differences to the WT locus to accurately quantify editing efficiency (*Figure 1—figure supplement 2*, Materials and methods). Testing the multi-sgRNA approach across three loci revealed a shift in indel profiles toward large deletions and multiple indels with deletions of sequences between sgRNA cut sites as well as a modest increase in editing efficiency (*Figure 1e*, *Figure 1—figure supplement 1d,e*). Cas9 RNPs assembled with sgRNAs rather than with crRNA:tracrRNA duplexes mediated higher editing efficiency (*Figure 1—figure supplement 1d*). Through an expansive grid search of electroporation conditions, we next identified a condition (P3, DJ108) with editing efficiencies >90%, high cell recovery, and high specificity in detecting loss of TNF-α secretion upon stimulation with LPS from *E. coli* O55 (*E. coli* LPS) after knockout of *TNF* or the LPS receptor *TLR4* (*Poltorak et al., 1998*) but not *CXCR4*, a chemokine receptor not involved in LPS signaling (*Figure 1—figure supplement 3a*). We used this condition for all further experiments, although other conditions also permitted efficient genome editing (*Figure 1—figure supplement 3a*). Benchmarking the procedure for 10 genes in moDCs derived from three independent donors revealed editing efficiencies >80% for all genes and >90% for most genes (*Figure 1f*, *Figure 1—figure supplement 3b*).

In parallel, we challenged knockout moDCs from two donors with *E. coli* LPS, a TLR4 agonist, and measured production of two cytokines: (i) TNF-α, which is induced by MYD88 and TRIF (*TICAM1*) downstream of TLR4, and (ii) CXCL10 (IP-10), which is induced by TRIF via activation of IRF3 and production of interferon beta (*IFNB1*), independently of MYD88 (*Fitzgerald et al., 2003*; *Fitzgerald and Kagan, 2020*; *Yamamoto et al., 2003*; *Yamamoto et al., 2002*) (an overview of TLR4 signaling pathways is included in *Figure 1—figure supplement 4*). We normalized cytokine production for each knockout population to that from moDCs with knockout of *RPE65*, a retinal pigment epithelium-specific gene that does not contribute to DC function and serves as a neutral control. *TLR4* knockout abolished production of both TNF-α and CXCL10, knockout of *MYD88* or *TNF* reduced TNF-α production (and mildly increased CXCL10 production in at least one donor), *TICAM1* knockout strongly reduced CXCL10 production and moderately reduced TNF-α production, and knockout of *IFNB1* or *IRF3* only reduced CXCL10 production (*Figure 1g*, *Figure 1—figure supplement 3c*). The effect of knocking out *IRF3* was weak, perhaps due to redundancy with other transcription factors such as *IRF7* or due to incomplete protein depletion. Separately, knockout moDCs from an independent donor challenged with *E. coli* LPS or Pam3CSK4, a TLR2/TLR1 agonist, responded as expected: knockout of *TNF* or *MYD88* reduced the response to both stimuli, whereas knockout of *TLR4* or *TLR2* only reduced the responses to their cognate ligands (*Figure 1—figure supplement 3b*). Thus, our moDC genome editing strategy enables the detection of functional consequences of knockouts, demonstrating that we can effect protein depletion without perturbing the ability of moDCs to respond to innate immune stimuli.

To evaluate if our editing strategy leads to nonspecific changes in DC state, we measured expression levels of the markers CD11c, HLA-DR, CD83, and TLR4 as well as of B2M in knockout moDCs and unedited control moDCs by antibody staining. Staining patterns were qualitatively indistinguishable for moDCs with knockout of *B2M* or *RPE65*, moDCs electroporated with a non-targeting control sgRNA, or unedited moDCs (no electroporation/no RNP control) that had been cultured alongside the knockout moDCs in 96-well plates (*Figure 1—figure supplement 5*). Both knockout

and unedited moDCs from 96-well plates had mildly elevated levels of HLA-DR and B2M compared to unedited moDCs that had been cultured in flasks, indicating that culture conditions can affect DC state independent of editing. Although our antibody panel was not designed for precise measurements of B2M levels, B2M staining was reduced 72% and 65% in *B2M* knockout moDCs for the two donors, consistent with protein depletion. Together, these data provide evidence that our editing strategy does not perturb DC state.

## DC responses to LPSs are specific to bacterial species and vary across individuals

We next sought to apply our genome editing strategy to investigate a central question in innate immunity. Despite LPS being a classic inflammatory molecule, humans are colonized by trillions of Gram-negative microbes that generate milligram to gram quantities of LPS in the intestinal tracts without tonic inflammation. This observation has been attributed in part to the different chemical structures and immunomodulatory activities of LPSs from commensal Bacteroidetes, among the most common Gram-negative phyla in the guts of western individuals (*Wexler and Goodman, 2017*), compared to the canonical inflammatory LPSs from *E. coli* and related Proteobacteria (*Coats et al., 2011*; *Tan et al., 2015*; *Vatanen et al., 2016*; *d'Hennezel et al., 2017*). Indeed, the Bacteroidetes-to-Proteobacteria LPS ratio in the gut microbiome has been associated with the incidence of type 1 diabetes (*Vatanen et al., 2016*), suggesting that LPSs from the human microbiome contribute to shaping immune function. The biological activities of LPSs from gut Bacteroidetes, however, have remained controversial as they have been reported to be both TLR4 antagonists and agonists (*d'Hennezel et al., 2017*; *Steimle et al., 2019*; *Vatanen et al., 2016*). We set out to establish how human DCs respond to LPS from Bacteroidetes and more broadly how DCs discriminate different LPSs and initiate specific immune responses.

We focused on LPS from *B. theta*, an abundant member of the human gut microbiota whose LPS biosynthetic machinery has been characterized, allowing us to genetically manipulate its structure (*Coats et al., 2011*; *Cullen et al., 2015*; *Jacobson et al., 2018*). (*B. theta* LPS formally is a lipooligosaccharide, but we refer to it as LPS for clarity.) We purified LPS from a *B. theta* strain carrying deletions of all eight capsular polysaccharide (CPS) biosynthetic gene clusters (*Porter et al., 2017*; *Rogers et al., 2013*) to obtain LPS without other contaminating glycolipids, hereafter referred to as *B. theta* WT LPS (ΔCPS) or just *B. theta* LPS. Human moDCs stimulated with *B. theta* LPS secreted moderate levels of TNF-α as quantified by ELISA; this response was weaker than that elicited by *E. coli* LPS both in magnitude and apparent $EC_{50}$ but substantially stronger than that elicited by *Rhodobacter sphaeroides* LPS, a well-characterized TLR4 antagonist (*Figure 2a*; *Golenbock et al., 1991*). (We note that we cannot calculate $EC_{50}$'s in terms of molar concentrations due to the heterogeneity of LPS molecules. Nonetheless, because *E. coli* O55 LPS contains an O-antigen and thus has a greater average molecular weight than *B. theta* WT LPS, the trends we observe hold at the level of molar concentrations.) Although the pattern was consistent across moDCs from independent donors, response magnitude and $EC_{50}$ varied by sixfold and >20-fold, respectively (*Figure 2—figure supplement 1a*), even for moDCs processed in parallel, suggesting that donor-specific factors shape immune responses. Analysis of the transcriptional responses of moDCs by RT-qPCR and RNA-seq confirmed that *B. theta* LPS activated both MYD88 and TRIF signaling more weakly than *E. coli* LPS, with a more pronounced difference for TRIF signaling, again with donor-to-donor variation (*Figure 2b*, *Figure 2—figure supplement 1b,c*).

To further establish if *B. theta* LPS is a partial agonist of TLR4 rather than an antagonist, we turned to genetic engineering of *B. theta* LPS. LPSs of Bacteroidetes generally contain pentaacylated, monophosphorylated lipid A as opposed to the hexaacylated, diphosphorylated lipid A of Proteobacteria (*Coats et al., 2011*; *Weintraub et al., 1989*), in addition to other differences including lipid A acyl group structures and LPS glycan composition. A previous study had established that *B. theta* LPS modified to contain pentaacylated, diphosphorylated lipid A has increased capacity to stimulate TLR4 signaling via the endocytotic pathway (*Tan et al., 2015*). Hypothesizing that *B. theta* lipid A further lacking an acyl group would resemble known TLR4 antagonists (*Golenbock et al., 1991*) and thus have decreased immunostimulatory activity, we stimulated moDCs with LPS purified from a *B. theta* strain genetically engineered to produce tetraacylated, diphosphorylated lipid A (*B. theta* 4PP LPS) (*Jacobson et al., 2018*). *B. theta* 4PP LPS elicited substantially lower levels of TNF-α production and smaller transcriptional responses than *B. theta* WT LPS, with barely detectable

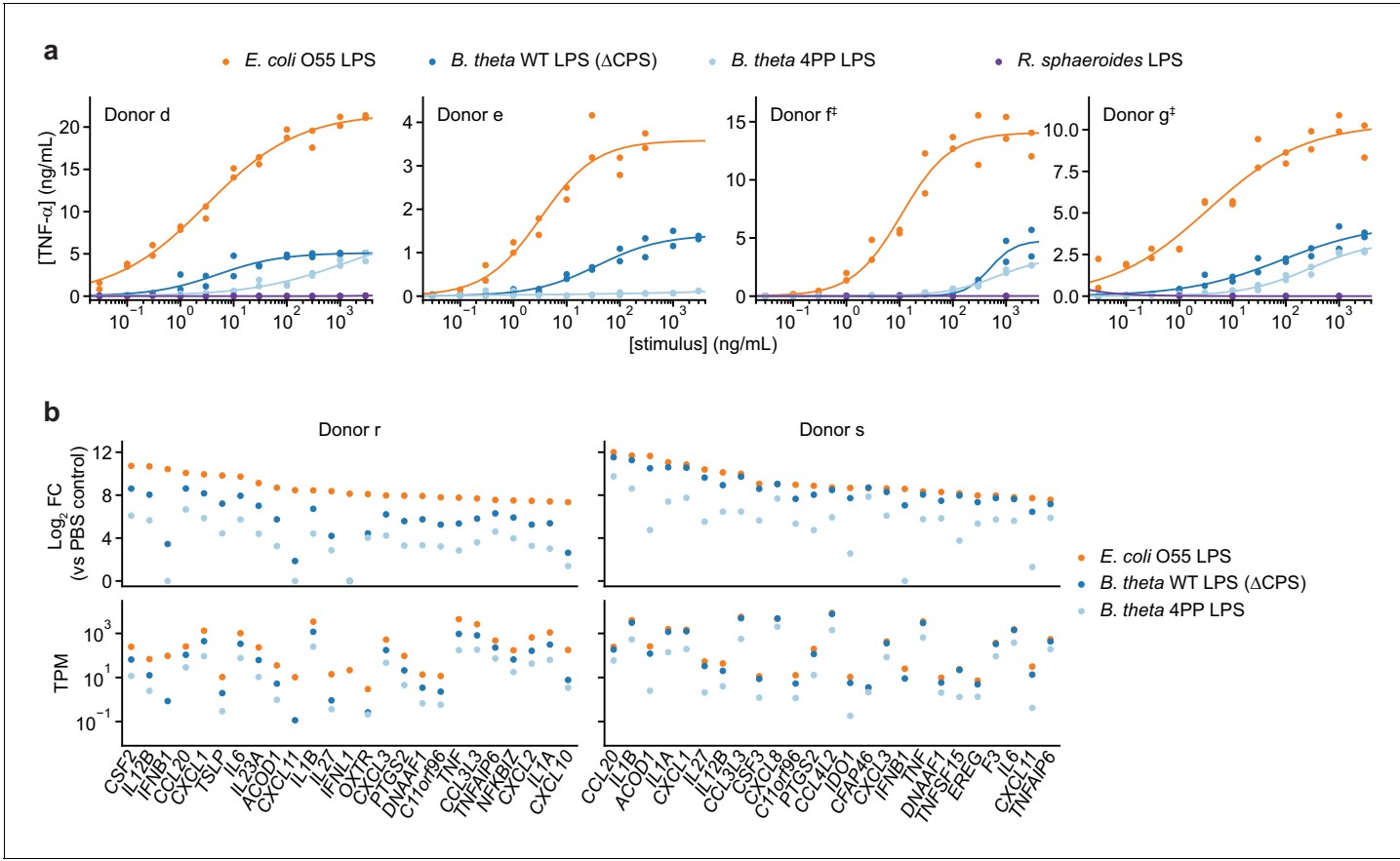

**Figure 2.** Responses of human moDCs to LPSs are specific to bacterial species and human donor. (a) TNF-α secretion after stimulation of moDCs from four independent donors with titration series of the indicated LPSs. Cells from donor e were not treated with *R. sphaeroides* LPS. Each data point represents an independent treatment of 20,000 moDCs. Lines denote a Hill curve fit. ‡ indicates moDCs that were processed in parallel (donors f and g). (b) Expression levels of selected genes after stimulation of moDCs from two donors with 10 ng/mL *E. coli* O55 LPS, 100 ng/mL *B. theta* WT LPS, or 100 ng/mL *B. theta* 4PP LPS, as determined by RNA-seq. Log₂ fold-changes compared to PBS-treated control cells or transcript counts per million are shown for the 25 protein-coding genes with the largest log₂ fold-changes after treatment with *E. coli* O55 LPS. *B. theta* WT LPS elicits weaker upregulation of genes than *E. coli* O55 LPS, with a more pronounced difference for genes downstream of TRIF such as *IFNB1*. *B. theta* 4PP LPS elicits even weaker upregulation of genes. Data represent means obtained from three independent treatment replicates for each treatment and donor. See also *Figure 2—figure supplement 1*, *Figure 2—figure supplement 2*, and *Figure 2—figure supplement 3*.

The online version of this article includes the following figure supplement(s) for figure 2:

**Figure supplement 1.** *B. theta* LPS and *E. coli* LPS elicit different responses from moDCs with responses that vary by donor.

**Figure supplement 2.** *TLR4* knockout does not completely eliminate the response to *B. theta* LPS.

**Figure supplement 3.** Comparison of transcriptonal profiles of edited and unedited moDCs.

responses in some donors (*Figure 2a,b*, *Figure 2—figure supplement 1c*). (Note that we purified *B. theta* 4PP LPS from a CPS-producing *B. theta* strain; we nonetheless attribute the difference to the lipid A modification because in preliminary work we observed similar qualitative differences when comparing *B. theta* 4PP LPS to *B. theta* WT LPS purified from a CPS-producing strain.) Thus, *B. theta* LPS is a partial TLR4 agonist whose immunostimulatory activity can be tuned by rational engineering.

To determine if the difference in activity between *B. theta* LPS and *E. coli* LPS arises solely from different capacity to activate TLR4, we tested how *TLR4* knockout affects responses to these two LPSs. *TLR4* knockout moDCs did not secrete detectable amounts of TNF-α in response to *E. coli* LPS but secreted substantial amounts of TNF-α in response to *B. theta* LPS, corresponding to 30–50% of the levels secreted by moDCs with knockout of the neutral control gene *RPE65* (*Figure 2—figure supplement 2a–c*).

We next assessed the transcriptional responses of moDCs with knockout of *TLR4* or *RPE65* (neutral control) to treatment with PBS (mock), *B. theta* WT LPS, or *E. coli* O55 LPS by RNA-seq (*Figure 2—figure supplement 3*). First, to further assess if our genome editing procedure perturbs DC state, we compared the transcriptional profiles of mock-treated moDCs with knockout of *RPE65* from two independent donors from this experiment (donors p and q) to the profiles of unedited moDCs from a previous experiment (donors r and s). Transcriptional profiles segregated primarily by donor, and each of the profiles of knockout moDCs more closely resembled those of unedited moDCs from one donor than the profiles of unedited moDCs from different donors resembled each other (*Figure 2—figure supplement 3a*). Knockout moDCs also had similar expression levels for a panel of DC and myeloid cell markers (*Figure 2—figure supplement 3b*). These data further suggest that our genome editing procedure does not perturb DC state.

Next, we assessed how *TLR4* knockout cells responded to *B. theta* WT LPS and *E. coli* O55 LPS. Whereas *TLR4* knockout abolished the response to *E. coli* O55 LPS, *TLR4* knockout moDCs retained a partial transcriptional response to *B. theta* WT LPS corresponding to activation of MYD88, but not TRIF (*Figure 2—figure supplement 2d,e*). These results suggest that receptors other than TLR4 contribute to recognition of *B. theta* LPS.

We confirmed that transcript levels were reduced in knockout cells, and transcript coverage was further reduced between sgRNA cut sites, indicating that the structures of most remaining transcripts were disrupted (*Figure 2—figure supplement 3c*). We note that transcript levels are generally not predictive of protein levels in CRISPR-edited cells (*Smits et al., 2019*), and thus these analyses only confirm that a necessary condition for TLR4 depletion is satisfied but are not sufficient to infer TLR4 depletion. Nonetheless, our observations of reduced TLR4 signaling in *TLR4* knockout cells (*Figure 2—figure supplement 2e*) suggest that the protein was also depleted.

## A genetic screen implicates receptors for *B. theta* LPS and drivers of inter-individual variation

To identify additional factors that contribute to recognition of *B. theta* LPS, we leveraged our moDC editing strategy to conduct an arrayed genetic screen (*Figure 3a*). We designed a focused library targeting ~300 genes including known and predicted pattern recognition receptors and multiple nodes of signaling pathways downstream of each receptor class (Materials and methods). We targeted each gene with two to three sgRNAs whenever multiple unique sgRNAs could be designed and distributed the sgRNAs over four 96-well plates, each of which also included four types of controls (*Figure 3—figure supplement 1*, Materials and methods). After electroporating moDCs from two independent donors with this library, we assayed editing efficiency and TNF-α secretion in response to 100 ng/mL *B. theta* WT LPS (Materials and methods, *Supplementary files 3*, *4*). Editing efficiency was high in both donors, with median efficiencies of 94.2% and 97.6% (*Figure 3b*, *Figure 3—figure supplements 2–4*). TNF-α secretion was generally unaffected for genes with editing efficiency <75%, suggesting a low false-positive rate (*Figure 3—figure supplement 4b*); we did not exclude such genes from further analysis. We identified 37 genes for which knockout altered TNF-α secretion for one donor ('donor h') and 41 genes for the second donor ('donor i'), with altered secretion defined as an absolute $\log_2$ fold-change in TNF-α levels ≥2 standard deviations from 0 (*Figure 3c*, *Figure 3—figure supplement 5a–c*, standard deviations calculated based on the $\log_2$ fold-changes for all neutral genes, see Materials and methods for details). For each donor, we observed altered TNF-α secretion for all four wells targeting the positive control gene *TNF* and one well targeting a neutral control gene (of 36 such wells).

Knockout of 15 genes (including *TNF*) impacted TNF-α secretion in response to *B. theta* WT LPS in both donors. Knockouts in the TLR4 signaling pathway strongly reduced TNF-α secretion, including *TLR4* and its co-receptor MD2 (*LY96*), *CD14* (which delivers LPS to TLR4 and initiates TLR4 endocytosis [*Zanoni et al., 2011*]), as well as genes involved in MYD88 signaling, the branch of TRIF signaling involved in NF-κB activation, and the NF-κB factor *RELA* (*Figure 3c*). Indeed, 6 and 7 of the 10 targeted genes in MYD88 signaling reduced TNF-α secretion in the two donors, with knockout of most of the remaining genes causing a less substantial reduction in TNF-α secretion (*Figure 3c*). Other knockouts also caused expected phenotypes; for example, moDCs with knockout of A20 (*TNFAIP3*), which inhibits LPS- and TNF-α-induced NF-κB signaling, secreted more TNF-α (*Figure 3c*, *Figure 3—figure supplement 5c*). Thus, our genetic screen accurately captured the genetic requirements for the response to *B. theta* LPS.

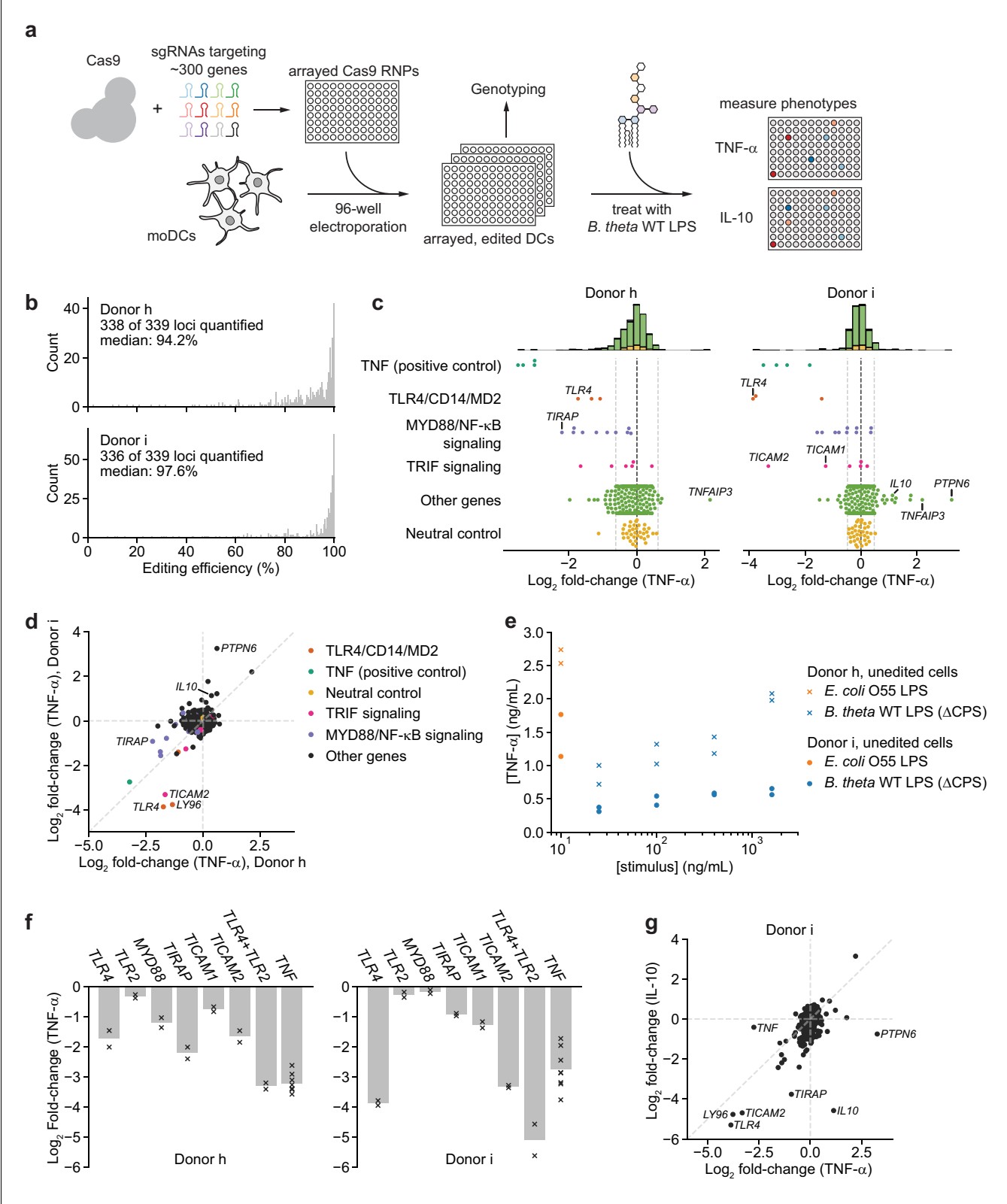

**Figure 3.** An arrayed genetic screen reveals how moDCs recognize LPS from a human gut bacterium and suggests mechanisms of inter-individual variation. (a) Schematic of genetic screen. (b) Observed editing efficiencies for loci targeted in the screen. Efficiencies were not obtained for one locus for donor h and three loci for donor i due to PCR failures; no locus failed for both donors. (c) TNF-α secretion of knockout moDC populations from two independent donors after stimulation with 100 ng/mL *B. theta* WT LPS, displayed as log₂ fold-changes compared to neutral controls within each of the *Figure 3 continued on next page*

*Figure 3 continued*

four 96-well plates and normalized to cell counts. Each data point represents the mean of two treatment replicates and two cell count replicates. Vertical dashed lines denote mean and two standard deviations of the phenotypes from all neutral gene controls. Distributions of all phenotypes are plotted in the stacked histograms, colored by category, at the top. (d) Comparison of TNF-α secretion from (c) for the two donors. (e) TNF-α secretion after stimulation of unedited moDCs (from no pulse/no RNP wells) from both donors with different concentrations of the indicated LPSs. Each data point represents an independent treatment. (f) TNF-α secretion for selected moDC knockout populations including moDCs with simultaneous knockout of *TLR4* and *TLR2* after stimulation with 100 ng/mL *B. theta* WT LPS. Data are shown as individual measurements (×) and mean of all treatment replicates (bars). (g) Comparison of TNF-α and IL-10 secretion from knockout moDC populations for moDCs derived from donor i after stimulation with 100 ng/mL *B. theta* WT LPS. Each data point represents the mean of two treatment replicates and two cell count replicates for TNF-α secretion and data from a single treatment replicate and two cell count replicates for IL-10 secretion. See also *Figure 3—figure supplement 1*, *Figure 3—figure supplement 2*, *Figure 3—figure supplement 3*, *Figure 3—figure supplement 4*, *Figure 3—figure supplement 5*, and *Figure 3—figure supplement 6*.

The online version of this article includes the following figure supplement(s) for figure 3:

**Figure supplement 1.** Layout of sgRNAs in the arrayed genetic screen.
**Figure supplement 2.** Percentage of genomic DNA reads assigned to different classes of outcomes after targeting each locus in the arrayed genetic screen (donor h).
**Figure supplement 3.** Percentage of genomic DNA reads assigned to different classes of outcomes after targeting each locus in the arrayed genetic screen (donor i).
**Figure supplement 4.** Further analysis of editing outcomes from genetic screens.
**Figure supplement 5.** Screen phenotypes.
**Figure supplement 6.** Validation of screen results in moDCs from two additional, independent donors.

Although the results for the two donors were similar overall, we noticed several key differences. Most prominently, knockouts of *PTPN6* (SHP-1) and to a lesser extent *IL10* increased TNF-α secretion in response to *B. theta* WT LPS in one donor ('donor i') but not the other ('donor h'), suggesting that these factors can suppress TNF-α secretion in response to LPS stimulation in a manner that differs among individuals (*Figure 3d*). Indeed, unedited moDCs from donor i secreted less TNF-α in response to both *B. theta* WT LPS and *E. coli* LPS than those from donor h (*Figure 3e*). Second, MYD88 signaling contributed particularly strongly to the response to *B. theta* WT LPS for donor h: knockout of *TIRAP*, the TLR4-proximal adapter for MYD88 signaling (*Fitzgerald et al., 2001*; *Fitzgerald and Kagan, 2020*), induced the strongest decrease in TNF-α secretion other than *TNF* itself. For donor i, *TLR4* and the TRIF pathway contributed more strongly to the response, as evidenced by strong decreases in TNF-α secretion upon knockout of *TLR4* alone, TRAM (*TICAM2*), and TRIF (*TICAM1*) (*Figure 3c,d*). These differences account for many of the donor-specific hits (*Figure 3d*, *Figure 3—figure supplement 5c*). A separate 40-gene validation experiment with cells from two additional, independent donors recapitulated these results, with results from each of the two validation donors aligning more closely with those from one of the initial donors (*Figure 3—figure supplement 6*).

We further investigated two specific observations. First, because *TIRAP* knockout caused a larger decrease in TNF-α secretion than *TLR4* knockout in donor h, we wondered if other TLRs contributed to the response to *B. theta* WT LPS. We focused on TLR2, canonically known as a receptor for lipopeptides and teichoic acids, because *TLR2* knockout caused the next-strongest decrease in TNF-α secretion among TLRs and because TLR2 has been implicated in the response to non-proteobacterial LPSs (*Di Lorenzo et al., 2020*; *Werts et al., 2001*), although these claims remain controversial. Indeed, moDCs with simultaneous knockout of *TLR4* and *TLR2* exhibited the strongest decreases in TNF-α secretion among all samples for both donors (*Figure 3f*, *Figure 3—figure supplement 4d*). In addition, knockout of *TLR2* alone reduced the response to *B. theta* WT LPS but not to *E. coli* LPS in a donor in our validation experiment (*Figure 3—figure supplement 6b*). These results are consistent with the possibility that TLR2 contributes to the response to *B. theta* LPS, although we cannot rule out the presence of contaminating lipopeptides in our LPS preparation.

Second, to analyze the interplay between PTPN6 and IL-10 in suppressing TNF-α secretion in moDCs from donor i, we measured IL-10 levels in the same samples. The effects of knockouts on TNF-α and IL-10 secretion were well correlated, suggesting that TLR4 signaling via the TRIF and MYD88 branches accounts for secretion of both TNF-α and IL-10. PTPN6 stood out as an exception: whereas *PTPN6* knockout strongly increased TNF-α secretion, it moderately decreased IL-10 secretion (*Figure 3g*), suggesting either that IL-10 acts upstream of PTPN6 in suppressing TNF-α

secretion or that PTPN6 specifically inhibits production of TNF-α and not IL-10. More broadly, these results demonstrate how combining our moDC editing strategy with multiple readouts can increase the resolution in evaluating immune response pathways.

## Discussion

In summary, we describe a CRISPR strategy to introduce knockouts and conduct genetic screens in DCs derived from human donors. Our strategy has four main strengths. First, our strategy is efficient and consistent, with median knockout efficiencies of >94% across >300 genes in two independent donors and comparable efficiencies at smaller scales across >10 further donors. Second, the knockouts are introduced directly in differentiated DCs, reducing the potential for batch effects that may occur when introducing knockouts in precursor populations prior to differentiation. As a corollary, any gene is in principle targetable in our strategy, including for example genes required for DC differentiation. Third, our strategy is compatible with a variety of readouts to interrogate the phenotypes of knockout moDCs, including cytokine profiling, RNA-seq, flow cytometry, microscopy, and co-culture assays, enabling dissection of DC biology at high resolution. Fourth, as a result of the high efficiency and elimination of batch effects, our strategy enables genetic screens at previously intractable scales of hundreds of genes and would readily scale further with access to automation.

Like any method, our strategy also bears limitations. First, because human DCs do not divide in culture, our strategy relies on degradation to deplete protein molecules produced before genome editing. We observe strong phenotypes after targeting immune receptors and few false negatives in the MYD88 pathway in our screen, suggesting that most proteins are depleted during the 5-day period after electroporation. Nonetheless, depleting exceptionally long-lived proteins would require prolonged culture, which may not be experimentally feasible. Second, as an RNP-based strategy, our strategy is not readily compatible with pooled screening because no sgRNA barcodes are inserted into the genome. This limitation has been circumvented in T-cells by combining lentiviral sgRNA delivery and Cas9 electroporation (*Shifrut et al., 2018*; *Ting et al., 2018*). Implementation of such a strategy in DCs is hampered by the recalcitrance of DCs to lentivirus infection and the tendency of DCs to undergo maturation when infection is forced, such as by co-transduction of Vpx, but may become feasible with the development of transduction strategies for DCs. Third, as for any CRISPR method, it is difficult to knock out highly homologous genes, such as some C-type lectin receptors, because of challenges in designing sgRNAs that are both active and specific. These limitations are important to consider in experimental design, and we demonstrate that in doing so the impacts of these limitations can be minimized.

Indeed, our genetic screen broadly recapitulates known LPS signaling pathways and further reveals two specific hypotheses regarding the recognition of *B. theta* LPS. First, we observe a potential contribution of TLR2 to signaling by *B. theta* LPS, as has also been proposed for *B. vulgatus* LPS (*Di Lorenzo et al., 2020*). This observation could result from either an intrinsic ability of *B. theta* LPS to bind both TLR4 and TLR2 or the presence of contaminating TLR2 ligands in our *B. theta* LPS preparation, which we have not ruled out. Second, we identify *PTPN6*/SHP-1 as a potential contributor to inter-individual variation in the response to LPS. Activation of *PTPN6*/SHP-1, in addition to regulating innate immunity, suppresses antigen cross-presentation (*Ding et al., 2016*), which some pathogens exploit to subvert adaptive immunity (*Khouili et al., 2020*). Further support for our finding of inter-individual variation in *PTPN6*/SHP-1 activity thus would have immediate implications both for understanding variations in pathogen susceptibility and for personalizing DC vaccines aimed at initiating CD8 T-cell responses.

Our observations of inter-individual variation in immune responses, alongside many previous reports (*Brodin and Davis, 2017*; *Fairfax et al., 2014*; *Sanz et al., 2018*; *Ye et al., 2014*), further underscore the importance of probing immune responses directly in donor-derived cells. For example, moDCs from different donors have distinct responses – both in magnitude and apparent affinity – to identical preparations of LPS. Such variation likely results from a combination of genetic factors and environmental exposures that together determine immune cell state, with contributions from *PTPN6*/SHP-1 activity emerging as a hypothesis from our data. Longitudinal studies with larger cohorts of donors will be required to dissect the relative contributions of these factors and to more clearly define genetic drivers of inter-individual variation, and our work now provides enabling tools for such studies.

Finally, we find that human DCs initiate specific responses to LPS from commensal Bacteroidetes: *B. theta* LPS induces an inflammatory response that is weaker in both magnitude and apparent $EC_{50}$ than the response induced by *E. coli* LPS and can be further dampened by rational modifications to the LPS lipid A portion. Despite donor-to-donor variation in the LPS response, these trends hold across all donors we examined. Our findings challenge the notion that *Bacteroides* LPSs are innocuous components of the human gut microbiota, as further evidenced by a recent report that homeostatic, TLR4-dependent induction of IFN-β by *B. fragilis* LPS contributes to antiviral immunity (*Stefan et al., 2020*). Variations in LPS structure across gut commensals instead alter capacity to activate TLR4 and may allow for engagement of new receptors altogether, with the potential for neomorphic activities as well as further complexity arising from combinatorial perception (*Antebi et al., 2017*). In this fashion, commensal LPSs likely contribute to shaping immune responses at the host-microbiome interface. Indeed, activation of TLRs by commensal immune ligands including LPS contributes to intestinal homeostasis (*Rakoff-Nahoum et al., 2004*). Understanding the underlying mechanisms, using for example the genetic approaches we describe, may in turn enable efforts to engineer LPSs with defined immunomodulatory capacities, akin to our *B. theta* 4PP LPS mutant.

Beyond LPS recognition, the availability of functional genomics tools for human DCs now opens the door to a range of applications including systematic functional genomics studies to dissect the roles of DC receptors and signaling pathways in mounting immune responses to commensals, pathogens, or tumor cells and targeted engineering of moDCs for therapeutic interventions such as neoantigen vaccination.

# Materials and methods

**Key resources table**

| Reagent type (species) or resource | Designation | Source or reference | Identifiers | Additional information |
|---|---|---|---|---|
| Strain, strain background (*Bacteroides thetaiotaomicron*) | VPI-5482 Δtdk ΔCPS | *Rogers et al., 2013* | Referred to as '*B. theta*' in the text | Acapsular mutant provided by Eric Martens's lab |
| Strain, strain background (*Bacteroides thetaiotaomicron*) | VPI-5482 Δtdk ΔBT1854 ΔBT2152 | *Jacobson et al., 2018* | Referred to as '*B. theta* 4PP' in the text | Mutant producing tetra-acylated, di-phosphorylated lipid A in a capsule-producing background |
| Biological sample (human) | PBMCs | AllCells | | Freshly isolated from de-identified healthy individuals and shipped overnight |
| Antibody | Anti-human CD14 (clone HCD14, mouse monoclonal), PE, PerCP-Cy5.5, or BV421 | BioLegend | PE: Cat#: 325605; RRID:AB_830678 PerCP-Cy5.5: Cat#:325621; RRID:AB_893252 BV421: 325627; RRID:AB_2561342 | Flow cytometry (2 µL per test) |
| Antibody | Anti-human CD80 (clone 2D10, mouse monoclonal), APC | BioLegend | Cat#: 305219; RRID:AB_2291403 | Flow cytometry (5 µL per test) |
| Antibody | Anti-human CD83 (clone HB15e, mouse monoclonal), APC-Cy7 | BioLegend | Cat#: 305329; RRID:AB_2566392 | Flow cytometry (4 µL per test) |
| Antibody | Anti-human CD86 (clone BU63, mouse monoclonal), FITC or BV605 | BioLegend | FITC: Cat#: 374203; RRID:AB_2721573 BV605: Cat#: 374213; RRID:AB_2734429 | Flow cytometry (5 µL per test) |

*Continued on next page*

*Continued*

| Reagent type (species) or resource | Designation | Source or reference | Identifiers | Additional information |
|---|---|---|---|---|
| Antibody | Anti-human HLA-DR (clone L243, mouse monoclonal), PE or FITC | BioLegend | PE: Cat#: 307605; RRID:AB_314683 FITC: Cat#: 307603; RRID:AB_314681 | Flow cytometry (5 µL per test) |
| Antibody | Anti-human CD11b (clone LM2, mouse monoclonal), PE-Cy7 | BioLegend | Cat#: 393103; RRID:AB_2734450 | Flow cytometry (5 µL per test) |
| Antibody | Anti-human CD11c (clone Bu15, mouse monoclonal), Pacific Blue, FITC, or PerCP-Cy5.5 | BioLegend | Pacific Blue: Cat#: 337212; RRID:AB_1595430 FITC: Cat#: 337213; RRID:AB_1877174 PerCP-Cy5.5: Cat#: 337209; RRID:AB_1279071 | Flow cytometry (5 µL per test) |
| Antibody | Anti-human CD205 (clone HD30, mouse monoclonal), PE | BioLegend | Cat#: 342203; RRID:AB_1626209 | Flow cytometry (5 µL per test) |
| Antibody | Anti-human B2M (clone 2M2, mouse monoclonal), PE | BioLegend | Cat#: 316306; RRID:AB_492839 | Flow cytometry (2 µL per test) |
| Antibody | Anti-human TLR4 (clone HTA125, mouse monoclonal), APC | BioLegend | Cat#: 312815; RRID:AB_2562486 | Flow cytometry (4 µL per test) |
| Sequence-based reagent | Purified sgRNAs | This paper/Synthego | Library available from Synthego as 'Pattern Recognition Receptors and Signaling Pathway arrayed library' | Sequences and genomic binding locations listed in *Supplementary file 1* |
| Sequence-based reagent | Genomic locus amplification primers | This paper | | Sequences listed in *Supplementary file 1* |
| Sequence-based reagent | qPCR primers against ACTB, IFNB1, TNF, CXCL10 | Universal Probe Library (Roche) | | Sequences included in Materials and methods section |
| Peptide, recombinant protein | Human GM-CSF | Gemini Bio | Cat#: 300–124P | Used at 50 ng/mL in DC differentiation medium |
| Peptide, recombinant protein | Human IL-4 | Gemini Bio | Cat#: 300–154P | Used at 20 ng/mL in DC differentiation medium |
| Peptide, recombinant protein | *Streptococcus pyogenes* Cas9 2xNLS | Synthego | Available via Synthego as an 'Add-On Product' | |
| Commercial assay or kit | EasySep human monocyte enrichment kit (with or without CD16 depletion) | Stemcell Technologies | With CD16 depletion: Cat# 19059 Without CD16 depletion: Cat# 19058 | |
| Commercial assay or kit | SimpleStep human TNF alpha ELISA kit | Abcam | Cat#: ab181421 | |
| Commercial assay or kit | SimpleStep human IP-10 ELISA kit | Abcam | Cat#: ab173194 | |
| Commercial assay or kit | SimpleStep human IL-10 ELISA kit | Abcam | Cat#: ab185986 | |
| Commercial assay or kit | Stranded mRNA prep ligation kit | Illumina | Cat#: 20040534/20040532 | |
| Chemical compound, drug | Ultrapure *E. coli* O55:B5 LPS | Invivogen | Cat#: TLRL-PB5LPS | |
| Chemical compound, drug | *Rhodobacter sphaeroides* LPS | Invivogen | Cat#: TLRL-RSLPS | |
| Chemical compound, drug | *Bacteroides thetaiotaomicron* WT LPS (ΔCPS) | *Jacobson et al., 2018* this paper | Referred to as 'B. theta LPS' or 'B. theta WT LPS (ΔCPS)' in the text | |

*Continued on next page*

*Continued*

| Reagent type (species) or resource | Designation | Source or reference | Identifiers | Additional information |
|---|---|---|---|---|
| Chemical compound, drug | *Bacteroides thetaiotaomicron* 4PP LPS | *Jacobson et al., 2018* this paper | Referred to as 'B. theta 4PP LPS' in the text | |
| Software, algorithm | Knock-knock v0.3 | https://github.com/jeffhussmann/knock-knock and *Canaj et al., 2019* | | |
| Other | Ghost Dye Violet 510 | Tonbio Biosciences | Cat#: 13–0870 T100 | Flow cytometry viability stain (0.1 µL per 100 µL cells) |

## Reagents

Complete RPMI medium was generated by supplementing RPMI 1640 medium containing 25 mM HEPES, 2 mM L-glutamine, 2 g/L NaHCO$_3$ (Gibco, Dublin, Ireland) with 10% (v/v) standard fetal bovine serum (VWR, Wayne, PA), 100 units/mL penicillin, 100 µg/mL streptomycin, and 2 mM L-glutamine (Gibco). Lyophilized recombinant human GM-CSF (Gemini Bio, Sacramento, CA) and recombinant human IL-4 (Gemini Bio) were reconstituted to 100 µg/mL and 40 µg/mL, respectively, in sterile ddH$_2$O, aliquoted into 40–100 µL aliquots, and frozen at –30°C until use. Fluorescently labeled antibodies against human CD14 (clone HCD14, PE-, PerCP-Cy5.5-, or BV421-labeled), CD80 (clone 2D10, APC-labeled), CD83 (clone HB15e, APC-Cy7-labeled), CD86 (clone BU63, FITC- or BV605-labeled), HLA-DR (clone L243, PE- or FITC-labeled), CD11b (clone LM2, PE-Cy7-labeled), CD11c (clone Bu15, Pacific Blue-, FITC-, or PerCP-Cy5.5-labeled), CD205/DEC205 (clone HD30, PE-labeled), B2M (clone 2M2, PE-labeled), and TLR4 (clone HTA125, APC-labeled) were obtained from BioLegend (San Diego, CA). Ghost Dye Violet 510 was obtained from Tonbo Biosciences (San Diego, CA). Ultrapure LPS from *E. coli* O55:B5 and *Rhodobacter sphaeroides*, along with Pam3CSK4, were obtained from Invivogen (San Diego, CA). Solid medium used for bacterial growth was BHI/blood agar, made from Brain Heart Infusion Agar (BD Biosciences, San Jose, CA) with 10% defibrinated horse blood (Hemostat Laboratories, Dixon, CA). Liquid medium used for bacterial growth was supplemented BHI broth, made by preparing 1 L Brain Heart Infusion Broth (BD), and immediately before starting cultures adding 1 mL bovine hemin stock (Sigma, St. Louis, MO), 5 mg/mL in 1 N sodium hydroxide and filter sterilized, and 10 mL L-cysteine hydrochloride (Sigma), 50 mg/mL in Milli-Q water and filter sterilized. Sources of sgRNAs and Cas9 are listed below.

## Bacterial culture

*B. theta* strains were stored at –80°C in growth medium mixed in equal volume with 50% glycerol in water. Strains were streaked from glycerol stocks onto BHI/blood agar using plastic inoculating loops. Strains were allowed to grow 24–48 hr in an anaerobic chamber. Single colonies were used to inoculate 4 10 mL aliquots of supplemented BHI broth per strain, and after 24 hr the 10 mL cultures were expanded to 1 L each in glass bottles, producing 4 L total culture volume per strain. Cultures were allowed to grow to stationary phase (24–36 hr) and were pelleted at 3400 × *g* for 1 hr at 4°C. Pellets were washed in PBS and shipped frozen to the UCSD GlycoAnalytics Core for LPS purification.

## *B. theta* strains

Both the acapsular *B. theta* strain (ΔCPS) and the *B. theta* 4PP strain have been previously reported (*Porter et al., 2017*; *Rogers et al., 2013*; *Jacobson et al., 2018*). Briefly, the mutants were created using homologous recombination, creating scarless knockouts of the target genes/gene clusters with no remaining antibiotic resistance markers. The acapsular strain has had all known *B. theta* capsular polysaccharide gene clusters deleted (eight clusters in total), and the 4PP strain has had only lipid A acyltransferase *BT2152* and lipid A phosphatase *BT1854* deleted, not the CPS gene clusters.

## LPS purification

*B. theta* LPS preparations were performed by Biswa P. Choudhury at the UCSD GlycoAnalytics Core. A cell pellet from 4 L confluent culture of each *B. theta* strain was suspended in Milli-Q water and mixed with an equal volume of 90% phenol solution (Sigma, 328111). The suspension was stirred continuously and maintained at 68°C ± 2°C for 30 min. After cooling in an ice bath, suspensions were centrifuged at 3500 rpm at 10°C for 45 min and the upper layer removed to a clean Falcon tube. The remaining layers were extracted again with an equal volume of water for 30 min, cooled, and centrifuged as before. The upper layers were pooled and dialyzed (1000 MWCO, regenerated cellulose tubing) against 4 L of water for 4 days, replacing the water twice per day. The dialysate was lyophilized, resuspended in water, and subjected to ultracentrifugation at 105,000 × *g* for 4 hr. The pellet was resuspended in water, treated with DNase I, RNase A, and proteinase K, followed by another round of ultracentrifugation as above. The resulting pellet was resuspended in water and lyophilized.

## Differentiation of monocyte-derived DCs

Human moDCs were differentiated from monocytes isolated from commercially sourced fresh peripheral blood mononuclear cells (PBMCs) from de-identified healthy human donors (AllCells, Alameda, CA). The authors did not obtain identifiable private information on donors. The commercial vendor obtained informed consent from all donors covering all experiments and data reported in this manuscript. Monocytes were isolated from PBMCs by negative magnetic selection using the EasySep human monocyte enrichment kit without CD16 depletion (StemCell) following the manufacturer's instructions and using a Big Easy magnet or Easy 50 magnet (StemCell Technologies, Vancouver, Canada). Enriched monocytes were generally >80% CD14-positive, as assessed by flow cytometry on an LSR-II flow cytometer (BD BioSciences) or an Attune NxT flow cytometer (Thermo Fisher Scientific, Waltham, MA). Cell counts were determined in duplicate using a Countess II automated hemocytometer (Thermo Fisher Scientific). The isolated monocytes were cultured in complete RPMI medium, supplemented with 50 ng/mL GM-CSF and 20 ng/mL IL-4 immediately prior to use, at a density of $1 \times 10^6$ to $1.3 \times 10^6$ per mL at 37°C and 5% $CO_2$ for 7 days. Medium was exchanged every 2 or 3 days during this period (twice total). moDCs on day 7 were generally positive for CD80, CD83, HLA-DR, CD11b, CD11c, and CD205 and expressed intermediate levels of CD86 and low to intermediate levels of CD14 with some donor-to-donor variation, as assessed by flow cytometry on an LSR-II flow cytometer (BD BioSciences) or an Attune NxT flow cytometer (Thermo Fisher Scientific). All manipulations were performed in polystyrene conical tubes.

All experiments reported in this manuscript were conducted using the methods described above. Preliminary experiments were also performed after isolation of monocytes using the EasySep human monocyte enrichment kit with CD16 depletion (StemCell Technologies) and the EasySep human monocyte isolation kit (StemCell Technologies) with equivalent results. Analogous experiments were also performed with cells cultured in RPMI 1640 medium without supplementation of penicillin/streptomycin/L-glutamine, with equivalent results. RNA-seq data from moDCs from the same donor differentiated in parallel with and without penicillin/streptomycin/L-glutamine were virtually identical (not shown).

## Harvesting of moDCs

For all assays, both non-attached and loosely attached moDCs were harvested and then combined. The culture supernatant containing the non-attached cells was first transferred to a conical tube. The remaining attached cells were then detached by addition of CellStripper (Corning, Corning, NY), a non-enzymatic dissociation solution, to the flask (3 mL for a T-150 flask, 1.5 mL for a T-75 flask, 0.5 mL for a T-25 flask) and incubation at 37°C and 5% $CO_2$ for 15 min. The cells were further detached by pipetting and gently tapping the flasks. The suspension was aspirated into a new conical tube and another round of detachment with CellStripper was performed for 5 min. The detached cells were combined, centrifuged at 100 × g for 10 min, resuspended in RPMI medium, and combined with the non-attached cells. Cell counts were determined in duplicate using a Countess II automated hemocytometer (Thermo Fisher Scientific); at least two squares were counted for each replicate. All manipulations were performed in polystyrene conical tubes.

## Treatments of moDCs

To prepare moDCs for treatments, an aliquot of cells containing an appropriate cell number was centrifuged at $100 \times g$ for 10 min. The cells were resuspended in complete RPMI medium without cytokines. For readout by ELISA, cells were dispensed into flat-bottom 96-well plates in aliquots of 20,000 cells in 200 μL and incubated at 37°C and 5% $CO_2$ for 2–4 hr. Each experiment contained medium-only (no cells) and PBS treatment (unstimulated/no treatment control) negative controls. For subsequent RNA isolation, cells were dispensed into flat-bottom 24-well plates in aliquots of 200,000–250,000 cells at $1 \times 10^6$ cells/mL, as indicated for each experiment, and incubated at 37°C and 5% $CO_2$ for 2–4 hr. To initiate the stimulation, purified LPS or PBS (no treatment control) was added to each well to the final desired concentration. LPS stocks were generally prepared at a $20\times$ concentration such that all wells received an equivalent volume of stimulant.

For readout by ELISA, the cells were incubated with the stimuli at 37°C and 5% $CO_2$ for 20 hr, at which point the supernatants were transferred into a V-bottom 96-well plate, centrifuged at $3200 \times g$ for 10 min to remove residual bacteria and cell debris, transferred to new plates, and frozen at –30°C in aliquots.

For RNA purifications, the cells were incubated with the stimuli at 37°C and 5% $CO_2$ for 2 hr. To harvest RNA from treated cells, a $3\times$ volume of TRIzol LS reagent (Ambion, Naugatuck, CT) or TRI Reagent (Zymo Research, Irvine, CA) was added directly to the cells. The suspension was mixed by pipetting to lyse the cells, followed by RNA isolation using the Direct-zol RNA Miniprep kit (Zymo Research) including an on-column DNase I digestion step. Purified RNA was quantified using a Qubit Fluorometer (Thermo Fisher Scientific) and stored at –80°C until use.

## Quantification of cytokine concentrations by ELISA

TNF-α concentrations in undiluted or appropriately diluted supernatants were determined by ELISA using the SimpleStep human TNF alpha ELISA kit (Abcam, Cambridge, MA), performed following the manufacturer's instructions and with endpoint absorbance measurements at 450 nm on an Infinite M200 Pro plate reader (Tecan, Männedorf, Switzerland). For each experiment, absorbance measurements from wells containing a twofold dilution series of purified TNF-α (31.25 pg/mL to 2000 pg/mL, in duplicate) were used to calculate a calibration curve using a four-parameter logistic fit, which in turn was used to calculate TNF-α concentrations in all sample wells. Concentrations of CXCL10 and IL-10 were determined equivalently using SimpleStep human IP-10 ELISA kit (Abcam) and the SimpleStep human IL-10 ELISA kit (Abcam), respectively, following the manufacturer's instructions. When handling multiple 96-well plates simultaneously, plates were staggered in 3-min intervals starting with the last wash step to ensure that incubation times with the development solution and stop solution were constant.

## RT-qPCR

To generate cDNA, purified RNA was reverse-transcribed using SuperScript III Reverse Transcriptase (Thermo Fisher Scientific) with oligo(dT) primers in the presence of RNaseOUT Recombinant Ribonuclease Inhibitor (Thermo Fisher Scientific) or using SuperScript IV VILO Master Mix (Thermo Fisher Scientific). All reactions in a given experiment were normalized to contain the same amount of RNA (250–600 ng depending on the experiment). cDNA was diluted 1:10 and stored at –30°C until use. qPCR was performed using the KAPA SYBR FAST qPCR Master Mix (Roche, Basel, Switzerland) in 20 μL reactions containing 3 μL diluted cDNA and 200 nM of each primer. Reactions were run on a LightCycler 480 Instrument (Roche). All reactions were performed in technical triplicates. RT-qPCR primers were chosen as intron-spanning primers, when possible, from the Universal ProbeLibrary (Roche), with the following sequences:

*ACTB*: GCTACGAGCTGCCTGACG (fw), GGCTGGAAGAGTGCCTCA (rv)
*IFNB1*: CTTTGCTATTTTCAGACAAGATTCA (fw), GCCAGGAGGTTCTCAACAAT (rv)
*TNF*: CAGCCTCTTCTCCTTCCTGAT (fw), GCCAGAGGGCTGATTAGAGA (rv)
*CXCL10*: GAAAGCAGTTAGCAAGGAAAGGT (fw), GACATATACTCCATGTAGGGAAGTGA (rv)

## Preparation of sequencing libraries for RNA-seq and data analysis

RNA-seq libraries were prepared from purified RNA using the Stranded mRNA Prep Ligation kit (Illumina, San Diego, CA) in 96-well format, following the manufacturer's instructions. Input RNA amounts were held constant for all samples for a given donor, between 300 and 600 ng per reaction depending on the experiment. Final libraries were validated and quantified using the 2100 Bioanalyzer (Agilent, Santa Clara, CA) using the High Sensitivity DNA kit (Agilent). Paired-end 100 or paired-end 150 sequencing was performed on a HiSeq 4000 (Illumina). Reads were aligned strand-specifically to the human genome (GRCh38) using the spliced read aligner STAR (*Dobin et al., 2013*), version 2.6.0, against an index containing features from Gencode release 34. Quantification of gene counts was carried out with featureCounts (*Liao et al., 2014*), version 1.6.2, using only uniquely mapped reads to the reverse strand. Differential expression analysis was carried out on gene counts using DESeq2 (*Love et al., 2014*), including only genes with an average count >2 across all conditions. Transcript counts per million were calculated by dividing gene counts by effective transcript length, using transcript length from Gencode annotations and an average fragment length of 160 (the expected fragment size from the kit and consistent with estimates determined by Kallisto [*Bray et al., 2016*]), followed by normalization to total transcript counts. All other analyses were performed in python3.6.

## sgRNA sequences

For initial experiments, an sgRNA sequence for *AAVS1* was chosen from a previous report (*Mali et al., 2013*) and sgRNA sequences for *TNF* and *TLR4* were chosen as the top predicted guides from the Brunello library (*Doench et al., 2016*). All other sgRNAs were purchased from or provided by Synthego (Menlo Park, CA), designed according to their multi-guide RNA strategy (*Stoner et al., 2019*). Briefly, two or three sgRNAs are bioinformatically designed to work in a cooperative manner to generate small, knockout-causing, fragment deletions in early exons. These fragment deletions are larger than standard indels generated from single guides. The genomic repair patterns from a multi-guide approach are highly predictable based on the guide-spacing and design constraints to limit off-targets, resulting in a higher probability protein knockout phenotype. For the genetic screen, a Pattern Recognition Receptors and Signaling Pathway arrayed library was provided by Synthego. All sgRNA sequences used in this manuscript are listed in *Supplementary file 1*.

## RNP assembly

RNPs were assembled by complexing purified recombinant Cas9 from *Streptococcus pyogenes* (Synthego) with chemically synthesized sgRNAs (Synthego). Lyophilized sgRNAs targeting each gene (individual or multiple sgRNAs) were resuspended to 100 µM (total sgRNA concentration) in RNase-free TE buffer (10 mM Tris, 1 mM EDTA, pH 8) for 15 min at 25°C or overnight at 4°C with intermittent vortexing. Prior to use, sgRNA stocks were diluted to 25 µM in RNase-free H$_2$O. Both stocks were stored at –30°C and freeze-thawed up to five times. To assemble RNP for electroporation of $4 \times 10^5$ cells, 50 pmol sgRNA and 20 pmol Cas9 were combined and diluted to 20 µL with nucleofection solution P1 or P3 (with supplement added, Lonza, Basel, Switzerland). The mixture was incubated at 25°C for 10 min or up to 2 hr and immediately used to electroporate moDCs. For double knockouts, 50 pmol of sgRNA against each gene and 40 pmol Cas9 were combined in a total volume of 23 µL.

For experiments with guide RNAs in the crRNA:tracrRNA format (*Figure 1—figure supplement 1d*), lyophilized crRNAs and tracrRNA (Integrated DNA Technologies, Coralville, Iowa) were resuspended to 400 µM and 100 µM, respectively, in RNase-free TE buffer. crRNA:tracrRNA duplexes were generated by mixing equimolar amounts of crRNA and tracrRNA at a final concentration of 50 µM each, incubating the mixture at 95°C for 5 min, and allowing the mixture to cool to room temperature on the benchtop. To assemble RNP for electroporation of $4 \times 10^5$ cells, 50 pmol crRNA:tracrRNA duplex and 20 pmol Cas9 v3 (Integrated DNA Technologies, stock diluted to 20 µM in sterile PBS) were combined and diluted to 5 µL in PBS, following the manufacturer's instructions.

## moDC genome editing by electroporation of Cas9 RNPs

Genome editing was performed by electroporation of moDCs with pre-formed Cas9 RNPs. moDCs were detached as described above. A suspension containing an appropriate number of moDCs (4 ×

$10^5$ cells per electroporation +5% excess) was transferred to a new conical tube and centrifuged at 90 × g for 10 min. The cells were resuspended in 1–5 mL PBS and centrifuged again at 90 × g for 10 min. For electroporation with RNPs assembled with sgRNAs (all experiments in this paper except for *Figure 1—figure supplement 1d*, right), the cells were resuspended in 5 µL nucleofection solution P1 or P3 (with supplement added, Lonza) per 4 × $10^5$ cells. Aliquots of 4 × $10^5$ cells were transferred into individual wells of 16-well or 96-well nucleofection cuvettes (Lonza), combined with 20 µL pre-formed RNP or nucleofector solution (no RNP control), and immediately electroporated using pulse code DJ-108 (solution P3) or other pulse codes, as described, using a Nucleofector-4D (Lonza) or a 96-well shuttle (Amaxa/Lonza) attached to a Nuclefector-4D. For electroporation with RNPs assembled with crRNA:tracrRNA duplexes *Figure 1—figure supplement 1d*, right, the cells were resuspended in 20 µL nucleofection solution P1 or P3 (with supplement added, Lonza) per 4 × $10^5$ cells and 5 µL pre-formed RNP or nucleofector solution (no RNP control) was added. For no electroporation control cells, cells were treated identically, except that the corresponding cuvette well was not subjected to an electroporation pulse. Immediately after electroporation, 75 µL pre-warmed complete RPMI medium supplemented with 50 ng/mL GM-CSF and 20 ng/mL IL-4 were added to each well without disturbing the cells by letting the medium run down the side of the cuvette. After incubation at 37°C and 5% $CO_2$ for 1 hr, the cells were mixed by pipetting and then split into two wells of a flat-bottom 96-well plate filled with 50 µL pre-warmed complete RPMI medium supplemented with 50 ng/mL GM-CSF and 20 ng/mL IL-4. The cells were incubated at 37°C and 5% $CO_2$ for 5 days, with medium replenished after 2 or 3 days and then used for assays.

The final electroporation condition (solution P3, pulse code DJ-108, 4 × $10^5$ cells electroporated with 10 pmol Cas9 and 25 pmol sgRNA) was obtained through iterative grid searches of different conditions. In the process, several other conditions were also found to yield good results, including nucleofection solution P1 with pulse code CB-128 (*Figure 1—figure supplement 3a*). In initial experiments, we also used solution P1 with pulse code CB-150 (*Figure 1c*, *Figure 1—figure supplement 1*). Reducing the amount of Cas9 RNP led to a reduction in editing efficiency.

## Phenotyping of edited moDCs and DNA extraction

For all experiments in this manuscript, electroporated moDCs were harvested for phenotyping and genotyping 5 days post-electroporation, with the exception of data presented in *Figure 1—figure supplement 1a*, for which some moDCs were harvested 3 days post-electroporation. Both non-attached and attached cells were harvested and then combined, largely as described above. Briefly, the culture supernatants containing non-attached cells were first transferred to V-bottom 96-well plates. The remaining attached cells were then detached by addition of 25 µL CellStripper solution (Corning) per well and incubation at 37°C and 5% $CO_2$ for 15 min. The cells were further detached by gentle pipetting and tapping of the plate and the suspension was combined with the cell supernatant. Another round of detachment with CellStripper was performed for 5 min and the suspension was combined with the suspension containing the remaining cells. The cell suspensions were centrifuged at 100 × g for 10 min and resuspended in 100 µL complete RPMI medium without cytokines. Cells with the same knockout were combined (cells from each electroporation had been split over two separate wells of a 96-well plate) and used for phenotyping and genotyping.

To determine the responses of cells to stimuli by ELISA, aliquots of cells were transferred into flat bottom 96-well plates, diluted to 200 µL with complete RPMI medium without cytokines, incubated at 37°C and 5% $CO_2$ for 2–3 hr, and stimulated as described above. Each stimulation was performed in duplicate. Supernatants from stimulated cells were harvested and used to measure TNF-α levels as described above.

For subsequent RNA isolation, aliquots containing 1 × $10^5$ to 2.5 × $10^5$ cells were transferred into flat-bottom 24-well plates, diluted to 250 µL with complete RPMI medium without cytokines, incubated at 37°C and 5% $CO_2$ for 2–3 hr, and stimulated as described above. Each stimulation was performed in duplicate. RNA was extracted from treated cells as described above.

During the incubation prior to stimulation, aliquots of the remaining cell suspension were used to determine cell counts for each sample using a CellTiterGlo luminescence assay (Promega, Madison, WI). Briefly, replicate aliquots of cells were transferred into an opaque flat-bottom 96-well plate, diluted to 100 µL, and incubated at 25°C for 15–30 min. After addition of an equal volume of CellTiterGlo solution to each well, the plates were placed on an orbital shaker for 2 min and then incubated at 25°C for 10 min. Finally, luminescence in each well was recorded using a GloMax

Multi + luminescence plate reader (Promega). For some experiments, luminescence measurements from wells containing known numbers of unedited moDCs, as determined using a Countess II automated hemocytometer (Thermo Fisher Scientific), were used to calculate cell numbers for each sample. TNF-α secretion for each sample was then normalized to cell numbers. For other experiments, TNF-α secretion was simply normalized to background-subtracted luminescence readings. In benchmark experiments, cell counts were also determined by flow cytometry on an LSR-II flow cytometer (BD Biosciences) equipped with a 96-well autosampler. Cell counts determined by flow cytometry and luminescence were well correlated (*Figure 1—figure supplement 1a*) and all further cell counts were determined by luminescence.

To isolate genomic DNA from each sample for genotyping, aliquots of harvested moDCs were transferred to a 96-well V-bottom plate, centrifuged at 300 × g for 10 min, and resuspended in 50 µL QuickExtract DNA extraction solution (LuciGen, Middleton, WI). The suspensions were transferred to 96-well PCR plates and incubated at 65°C for 20 min and then at 98°C for 5 min using a thermocycler. The extracted genomic DNA was stored at –30°C until use.

### Flow cytometry of edited moDCs

DCs were differentiated and edited as described above. For each condition, moDCs electroporated in 3 wells of a 16-well cuvette (each well containing $4 \times 10^5$ cells at the time of electroporation) were harvested as described above and combined. After setting aside cells for genomic DNA extraction, the remaining cells were used for antibody staining. In parallel, moDCs from the same donors that had not been subjected to the electroporation procedure and had instead been cultured in T-25 flasks were also harvested and stained. Stains were performed in 96-well V-bottom plates. Aliquots of $3 \times 10^5$ to $5 \times 10^5$ cells per well were first stained with the amine-reactive viability dye Ghost Dye Violet 510 (Tonbo) by washing the cells twice in PBS (without protein additives) followed by incubation with 0.1 µL dye in 100 µL PBS on ice for 30 min in the dark. The cells were then washed twice in PBS containing 10% FBS, resuspended in PBS containing 10% FBS, and stained in a total volume of 100 µL with an antibody mix containing the following antibodies at the indicated final concentrations: anti-CD11c (Pacific Blue-labeled, clone Bu15, 20 µg/mL), anti-HLA-DR (FITC-labeled, clone L243, 6 µg/mL), anti-CD83 (APC-Cy7-labeled, clone HB15e, 8 µg/mL), anti-B2M (PE-labeled, clone 2M2, 1 µg/mL), anti-TLR4 (APC-labeled, clone HTA125, 8 µg/mL). Fluorescence-minus-one controls for each of the five antibodies were included for unedited cells from each donor. Cells were incubated with the antibody mixes on ice for 30 min in the dark, washed three times in PBS containing 10% FBS, and finally resuspended in 200 µL PBS containing 10% FBS. To set compensation, an aliquot of $2 \times 10^5$ heat-killed moDCs (incubated at 65°C for 15 min) was stained with Ghost Dye Violet 510 in the same fashion and then mixed with $2 \times 10^5$ live moDCs and washed as described above, and 1:1 mixtures of CompBead anti-mouse Ig, κ beads (BD Biosciences) and CompBead negative control beads (BD Biosciences) were stained with each individual antibody at the same concentrations and washed as described above. Flow cytometry data were recorded on an Attune NxT flow cytometer (Thermo Fisher Scientific) and analyzed using FlowCytometryTools 0.5.0 (https://eyurtsev.github.io/FlowCytometryTools/) and python 3.6. The gating strategy is illustrated in *Figure 1—figure supplement 5b*.

### Genotyping of edited moDCs (measurements of editing outcomes)

Genomic regions surrounding each cut site were PCR-amplified using a two-step protocol, largely as described (*Leenay et al., 2019*). Briefly, primer pairs were designed for each cut site using Primer-BLAST (*Ye et al., 2012*) to amplify a 200- to 450-base pair region, ensuring that all cut sites targeted by the pooled sgRNAs as well as a 50-base pair flanking region on each side of the cut sites were included, with a targeted $T_m$ of 60°C. Constant adapters (forward: 5′-CTCTTTCCCTACACGACGCTCTTCCGATCT-3′; reverse: 5′-CTGGAGTTCAGACGTGTGCTCTTCCGATCT-3′) were appended to the designed primer pairs. First-round PCRs of targeted sites were performed in 96-well format using at least 4000 genomic copies for each sample, 0.5 µM of each primer, and Q5 Hot Start High-Fidelity 2X master mix (NEB, Ipswich, MA) and the following protocol: 98°C for 30 s; 35 cycles of 98°C for 10 s, 60°C for 30 s, and 72°C for 30 s; and a final extension at 72°C for 2 min. Products from the first PCR were diluted 1:100 in ddH$_2$O and subjected to a second round of PCR using the constant adapters as annealing sites, appending Illumina P5 and P7 adapters and two eight-base barcodes on

both ends that together uniquely identified each sample. Twelve cycles of PCR were performed using the same conditions described above. After the second PCR, all samples were pooled and the combined samples were purified using a 0.8× AMPure XP purification (Beckman Coulter, Brea, CA). Final libraries were validated and quantified using the 2100 Bioanalyzer (Agilent) using the High Sensitivity DNA kit (Agilent) and sequenced in a 600-cycle paired-end run on a MiSeq Instrument (Illumina) using MiSeq v3 Reagent kits.

Sequencing data of editing outcomes were analyzed and quantified using knock-knock v0.3 (https://github.com/jeffhussmann/knock-knock) (*Canaj et al., 2019*). For a few loci, some amplicons contained large deletions with boundaries >20 bp from an sgRNA cut site that were classified as 'large deletions' by knock-knock but likely instead reflect amplification of partially complementary fragments, given in particular the rare occurrence of large deletions with individual sgRNAs. To avoid overestimating editing efficiency, reads with at least one alignment boundary >40 bp from an sgRNA cut site or both alignment boundaries > 20 bp from sgRNA cut sites were reclassified into the 'malformed layout' category. For all experiments in which a gene was targeted with multiple sgRNAs, sequencing counts were adjusted by the size difference to the WT locus, using the following formula:

$$count_{corr} = count * 2^{-0.014356*(l_{WT}-l_{read})}$$

where *count* is the raw count, $l_{WT}$ the length in bp of the WT locus, and $l_{read}$ the length in bp of the edited locus. See below for a description of how the coefficient was derived. Results from outcome classification, after correction for size, for all experiments except for the screen are listed in *Supplementary file 2*. Results for the screen, after correction for size, are listed in *Supplementary file 4*.

## Empirical assessment of amplicon size bias

To measure how amplicon size affects amplification and sequencing efficiency in our genotyping approach, we subjected pools of purified amplicons of defined sizes to our sequencing library preparation protocol and determined the resulting sequencing counts (*Figure 1—figure supplement 2a*). Briefly, we designed the strategy based on the following criteria:

1. Amplicons are of defined sizes between ~150 bp and ~500 bp (the range of amplicon sizes in our experiments) and amplified by the same primer pair;
2. Sequencing library preparation protocol is analogous to that used for genotyping, including similar effective template concentration and presence of excess non-productive genomic DNA;
3. Amplicon abundance is measured before sequencing library preparation and compared to final sequencing counts to estimate amplification and sequencing efficiency.

Briefly, we generated five amplicons with final lengths of 146 bp, 249 bp, 349 bp, 447 bp, and 539 bp and with constant annealing sites at the ends by PCR-amplifying different fragments of a gene encoding BFP with a constant forward primer and reverse primers positioned at the appropriate distances. Forward and reverse primers contained overhangs (identical for all reverse primers) to create annealing sites for sequencing library preparation. Following the PCR, each individual amplicon was gel-purified and quantified using a Qubit Fluorometer (Thermo Fisher Scientific). The five amplicons were then mixed into pools at 11 different molar ratios. For increased accuracy, the abundance of each fragment in these pools was measured using the 2100 Bioanalyzer (Agilent) using the High Sensitivity DNA kit (Agilent). Each pool was then diluted to 33 fM (about 20,000 template molecules per µL, equivalent to genomic DNA isolated from 10,000 cells) and 1 µL of diluted pool was used as template for the first-round PCR as described above, using a primer pair complementary to the constant overhangs on each fragment, designed with the same criteria as our other amplicon primers. The PCRs additionally contained 4 µL of genomic DNA from unedited DCs as excess non-productive template. The remainder of the sequencing library preparation was carried out as described above, with unique sequencing indices appended to each pool in the second-round PCR. The final libraries were sequenced on a MiSeq (Illumina) and counts for each fragment were determined by aligning reads to the expected amplicons.

To infer observation efficiency (amplification + sequencing) for each fragment, we reasoned that the starting and the final composition of the pool should be related by the specific observation

efficiency of each amplicon, which should shift the abundance of each fragment based on its specific observation efficiency:

$$m_i = \frac{e_i s_i}{\sum\limits_{j=1}^{n} e_j s_j}$$

where $m_i$ is the measured fractional abundance, $s_i$ the starting abundance, and $e_i$ the fragment-specific observation efficiency. Note that the equation takes this form because we can measure only fractional rather than absolute abundances of each amplicon at the end; thus

$$\sum\limits_{j=1}^{n} m_j = 1$$

To compute the efficiencies, we arbitrarily set the efficiency $e$ of the 447 bp fragment (which was included in all pools) to 1 ($e_n = 1$) and then solved the resulting linear equation system

$$m_i \sum\limits_{j=1}^{n} e_j s_j = e_i s_i$$

$$(m_i - 1) e_i s_i + m_i \sum\limits_{j \neq i}^{n-1} e_j s_j = -m_i s_n$$

to obtain the observation efficiencies (amplification + sequencing) $e_i$ for each fragment in each pool. Because we expected per-cycle PCR amplification efficiency to be a major contributor to these efficiencies, we compared $\log_2 e$ to fragment size and found it to be approximately linearly correlated (*Figure 1—figure supplement 2b*). We therefore estimated the contribution of each bp in size difference to observation efficiency using a linear regression of $\log_2 e$ against length in bp; we used the slope of this regression to correct sequencing counts as described above. We note that size bias appears to be less evident when small amplicons are already overrepresented in the input (*Figure 1—figure supplement 2b*), perhaps because under these conditions primers rather than nucleotides are the limiting component in PCR. Because our editing efficiencies are generally high and thus smaller fragments are more abundant at the outset than longer fragments, our correction approach is (intentionally) conservative and should not cause us to overestimate editing efficiency.

## RNA-seq of edited moDCs

RNA-seq libraries were prepared from purified RNA as described above. Input RNA amounts were held constant for all samples for a given donor (250 ng for donor p and 400 ng for donor q). For donor p, the RNA extraction for one replicate sample with knockout of *RPE65* and treated with 100 ng/mL *B. theta* LPS failed. Reads were aligned strand-specifically to the human genome, gene counts were quantified, and differential gene expression analysis was conducted as described above. Read coverage along transcripts was quantified using plastid (*Dunn and Weissman, 2016*). All other analyses were performed in python3.6.

## Design of library for arrayed genetic screen

To select genes to target in our arrayed genetic screen, we first included all genes from the following categories of pattern recognition receptors (PRRs): Toll-like receptors, NOD-like receptors, RIG-I-like receptors, C-type lectin receptors, Galectins, and SIGLECs. We then assembled a list of all genes encoding relevant signaling proteins downstream of these PRRs, including immediate adaptor proteins, kinases and ubiquitin ligases, the downstream transcription factors, as well as a limited subset of effector cytokines and cytokine receptors. Finally, we completed the gene list with additional genes of interest by surveying our RNA-seq data from human moDCs for expressed potential pattern recognition receptors such as predicted surface/membrane proteins, carbohydrate-binding proteins, and proteins containing a V-set domain using searches for Pfam domains, and by browsing the list of genes with the GO term 'innate immune response' that we had not yet included. This process ultimately resulted in a list of >400 genes. To narrow the list down to ~300 genes to enable

screening in four 96-well plates, we first eliminated pseudogenes and a few PRR-like genes with well-established functions. We then eliminated many genes involved in linear signaling pathways while ensuring that each pathway was targeted at multiple nodes in the final library. In total, we targeted 291 unique genes of interest.

The library additionally included four classes of controls: (1) non-targeting negative controls; (2) neutral controls (targeting negative controls); (3) targeting positive controls; and (4) essential controls. As non-targeting negative controls, we picked non-targeting negative controls #1 and #2 from Synthego. As neutral controls, we selected nine genes (*CRX*, *KCNV1*, *TRHR*, *LALBA*, *RPE65*, *F13B*, *OR2D2*, *OR51T1*, and *TAS2R9*) that are not expressed in moDCs, as assessed by our RNA-seq data and non-essential in any cell type surveyed at the time (from Project Achilles) (*Tsherniak et al., 2017*), and for which some functional annotation existed such as tissue-specific activity. These genes include olfactory and taste receptors as well as genes expressed only in specific tissues such as the retinal pigment epithelium or the testes. As a targeting positive control for TNF-α ELISA readout, we included *TNF*. As essential controls, we included the two core essential genes (*Hart et al., 2015*): *U2AF2* (a splicing factor) and *POLR2A* (a component of RNA polymerase II). In total, we included 14 controls. All sgRNA sequences are listed in *Supplementary file 1*.

Finally, all selected genes were arrayed into 96-well format, with the following design principles: each of the four 96-well plates contained each of the 14 controls in randomized positions on each plate; column 12 was left empty for no pulse electroporation controls and media-only ELISA controls; the remaining 74 positions on each 96-well plate were randomly filled with sgRNAs targeting genes of interest. The final plate layouts are depicted in *Figure 3—figure supplement 1* and listed in *Supplementary file 3*.

## Arrayed genetic screen

The Pattern Recognition Receptors and Signaling Pathway arrayed library targeting all selected genes with up to three sgRNAs per gene was provided by Synthego. For a few genes, high homology to other loci precluded selection of three unique sgRNAs within a 200 bp window; in those cases one or two sgRNAs were used. Purified, lyophilized sgRNAs were resuspended to 25 µM in 0.25× TE for 16 hr at 4°C, aliquotted into 96-well plates, and frozen at –80°C until use.

For each screen, monocytes were isolated from $1.5 \times 10^9$ PBMCs (AllCells) from a single healthy human donor and differentiated into moDCs as described above. Differentiated moDCs were electroporated with sgRNAs in 96-well format as described above. Plates were staggered in 10 min intervals to minimize the amount of time cells spent in nucleofection solution and the time delay between electroporation and addition of recovery media. On day 3 after electroporation, four wells of cells containing no pulse/no RNP control cells were harvested to assess responses of unedited cells to *B. theta* LPS and *E. coli* LPS and to determine an optimal *B. theta* LPS concentration for treatment of edited cells (*Figure 3e*). Maintenance, harvesting, and counting of electroporated moDCs were performed as described above. Plates were staggered for luminescence reads to keep incubation time with the luminescence substrate constant. All cells were treated with 100 ng/mL *B. theta* LPS in a volume of 200 µL. Concentrations of TNF-α and IL-10 in cell supernatants were determined by ELISA as described above. Plates were staggered in 7-min intervals throughout the entire process to keep incubation times constant.

Log$_2$ fold-changes in TNF-α or IL-10 secretion were calculated as follows:

1. ELISA absorbance values for each individual sample were background-corrected using absorbance values from media-only wells. For the IL-10 ELISAs, background-corrected absorbance below 0 (recorded for four samples) were assigned the background-corrected absorbance of the sample with the lowest value greater than 0.
2. Background-corrected absorbance values were normalized by the average luminescence (average of two replicate measurements) for each cell sample to calculate a cell count-normalized absorbance.
3. For each sample, the cell count-normalized absorbance was normalized to the median cell count-normalized absorbance of all nine neutral targeting controls on the same 96-well plate to calculate a fold-change in TNF-α or IL-10 secretion. Normalization was performed by plate to normalize for any plate effects.
4. Fold-changes were log$_2$ transformed to calculate log$_2$ fold-changes in TNF-α or IL-10 secretion.

5. Log$_2$ fold changes of replicate treatments were averaged to calculate the average log$_2$ fold change of each knockout population. Normality tests suggested that the log$_2$ fold-changes were generally normally distributed, rationalizing the averaging of the log$_2$-transformed values.

Raw and processed ELISA and cell count data are included in *Supplementary file 3*.

To assess editing efficiency at all loci, amplification primer design and sequencing library preparation were streamlined to increase throughput. A first round of amplification primers was designed using PrimerServer (*Zhu et al., 2017*), which uses a combination of primer3 and BLAST to predict amplification primers. Design criteria were: (1) amplification of a 200- to 450-base pair region, (2) inclusion of all cut sites targeted by the sgRNAs as well as a 35-base pair flanking region on each side of the cut sites was included, and (3) ideal T$_m$ of 60°C. Design criteria were successively relaxed if no primers matching these criteria were found, up to a maximum amplicon size of 500 base pairs and a minimum flanking distance of 25 base pairs. Primers containing overhangs as described above were ordered in 96-well format matching the sgRNA layout and tested for efficient amplification of the targeted locus by amplifying genomic DNA from unedited moDCs and sequencing the resulting amplicons on a MiSeq (Illumina), as described above. For loci with inefficient amplification or a high fraction of off-target amplicons, as assessed using knock-knock (*Canaj et al., 2019*), primers were designed using PrimerBLAST (*Ye et al., 2012*) as described above, and efficient amplification confirmed by sequencing. For some highly homologous locus pairs, such as *SFTPA1*/*SFTPA2* and *LGALS7*/*LGALS7B*, no primers could be designed that ensured completely unique amplification of each individual locus; the primers with the best discriminating power that fit all other design criteria (amplicon size, T$_m$, distance from cut site) were chosen. All primers are listed in *Supplementary file 1*.

Amplicon PCRs and sequencing sequencing library preparation were performed largely as described above, with the following modifications: (1) PCRs were performed in 384-well format; (2) first-round PCRs were set up using a Biomek FX liquid handling system with a 96-well head (Beckman Coulter); (3) first-round PCR products were diluted into Echo Qualified 384-Well Polypropylene Microplates using the Biomek FX; (4) PCR mastermix for the second-round PCR was dispensed into 384-well PCR plates using the Biomek FX; and (5) diluted first-round PCR products and indexing PCR primers were dispensed into the 384-well PCR plate using an Echo 525 acoustic liquid handler (Labcyte, San Jose, CA). Purification and validation of sequencing libraries, sequencing, classification of sequencing outcomes, and correction for amplicon size were performed as described above. For donor h, a small set of samples did not produce aligning sequencing reads in a first PCR attempt. These samples were repeated manually as described above, after which all but one sample produced aligning sequencing reads. Only successfully prepared samples were included for analysis. For donor i, the sequencing library preparation was repeated independently for >200 loci, which produced near-identical results (*Figure 3—figure supplement 4c*), validating that the sequencing library preparation strategy is robust and reproducible and that size-dependent amplification efficiency is consistent across PCRs. Samples with >100 size-corrected on-target reads (generally corresponding to >500 raw on-target reads) were included to estimate editing efficiencies. For some pairs of highly homologous loci (e.g. *SFTPA1* and *SFTPA2*), amplicons for both loci were detected with primer pairs designed to amplify each individual locus because it was impossible to design completely specific primer pairs with the criteria used. These amplicons were not excluded when calculating editing efficiency, such that editing efficiency is slightly underestimated for these loci. Results from outcome classification, after correction for size, are listed in *Supplementary file 4*.

## Sample sizes and sample size estimation

No sample-size calculation was performed in advance. All results were reproduced in cells from multiple independent donors, following conventions of the field. Within independent experiments, assays were performed in duplicate or triplicate following conventions of the field.

## Replication and data exclusion

All main findings were derived from experiments with cells from at least two independent donors. The main hits from the genetic screen were validated in cells from two additional, independent donors. All treatments were performed in duplicate for readout by ELISA and qPCR and in duplicate

or triplicate for readout by RNA-seq. Cell counts were generally conducted in duplicate. Information on number of replicates is contained in the figure legends.

For identification of differentially expressed genes in RNA-seq, only genes with an average count >2 across all conditions were included for analysis. Exclusion criteria for editing analysis are described in the corresponding methods sections.

### Code

Amplicon sequencing data were processed using the publicly available pipeline knock-knock (https://github.com/jeffhussmann/knock-knock) (*Canaj et al., 2019*). RNA-seq data were processed using STAR (*Dobin et al., 2013*), featureCounts (*Liao et al., 2014*), and DESeq2 (*Love et al., 2014*).

## Acknowledgements

We thank C Gross, J Hiatt, E Chow (all UCSF), A May (Chan-Zuckerberg Biohub), J Kagan (Boston Children's Hospital), J Chen (UT Southwestern), G Alberts (Lonza), and all members of the Weissman, Fischbach, and Gross labs for helpful discussions, and E Chow, D Martinez, and K Chaung from the UCSF Center for Advanced Technology for help with sequencing. The Pattern Recognition Receptors and Signaling Pathway arrayed library was provided by Synthego as part of a collaboration agreement. *B. theta* LPS purifications were performed by Biswa P Choudhury at the UC San Diego GlycoAnalytics Core. JSW is a Howard Hughes Medical Institute Investigator.

## Additional information

### Competing interests

Marco Jost: MJ consults for Maze Therapeutics. Jeffrey A Hussmann: JAH consults for Tessera Therapeutics. Michael A Fischbach: MAF is a co-founder and director of Federation Bio and Viralogic. Jonathan S Weissman: JSW consults for and holds equity in KSQ Therapeutics, Maze Therapeutics, and Tenaya Therapeutics. JSW is a venture partner at 5AM Ventures and a member of the Amgen Scientific Advisory Board. The other authors declare that no competing interests exist.

### Funding

| Funder | Grant reference number | Author |
|---|---|---|
| National Institutes of Health | K99 GM130964 | Marco Jost |
| National Institutes of Health | U01 CA217882 | Jonathan S Weissman |
| National Institutes of Health | DP1 DK113598 | Michael A Fischbach |
| National Institutes of Health | R01 DK11017404 | Michael A Fischbach |
| Chan Zuckerberg Initiative | | Jonathan S Weissman |
| Helmsley Charitable Trust | | Michael A Fischbach |
| Howard Hughes Medical Institute | Simons Faculty Scholar Award | Michael A Fischbach |
| Burroughs Wellcome Fund | Investigators in the Pathogenesis of Infectious Disease program | Michael A Fischbach |
| University of California, San Francisco | Postdoctoral Independent Research Grant | Marco Jost |
| Damon Runyon Cancer Research Foundation | DRG-2262-16 | Jeffrey A Hussmann |
| Howard Hughes Medical Institute | Investigator | Jonathan S Weissman |

The funders had no role in study design, data collection and interpretation, or the decision to submit the work for publication.

## Author contributions

Marco Jost, Conceptualization, Resources, Data curation, Formal analysis, Funding acquisition, Validation, Investigation, Visualization, Methodology, Writing - original draft, Project administration, Writing - review and editing; Amy N Jacobson, Conceptualization, Resources, Data curation, Formal analysis, Validation, Investigation, Visualization, Methodology, Writing - original draft, Project administration, Writing - review and editing; Jeffrey A Hussmann, Resources, Funding acquisition, Validation, Visualization, Methodology, Writing - review and editing; Giana Cirolia, Resources, Writing - review and editing; Michael A Fischbach, Jonathan S Weissman, Conceptualization, Supervision, Funding acquisition, Project administration, Writing - review and editing

## Author ORCIDs

Marco Jost (iD) https://orcid.org/0000-0002-1369-4908
Jonathan S Weissman (iD) https://orcid.org/0000-0003-2445-670X

## Decision letter and Author response

Decision letter https://doi.org/10.7554/eLife.65856.sa1
Author response https://doi.org/10.7554/eLife.65856.sa2

# Additional files

## Supplementary files

- Supplementary file 1. Sequences of sgRNAs and amplicon PCR primers used in this work, including details on sgRNA binding sites.

- Supplementary file 2. Counts corresponding to different outcomes, corrected for amplicon size, after amplicon sequencing for all donors except donors h and i.

- Supplementary file 3. Raw and processed TNF-$\alpha$ and IL-10 ELISA and cell count data for the genetic screens.

- Supplementary file 4. Counts corresponding to different outcomes, corrected for amplicon size, after amplicon sequencing for donors h and i.

- Transparent reporting form

## Data availability

Raw data from RNA-seq of unedited and edited moDCs are available at GEO under accession codes GSE161401 and GSE161466, respectively. Raw data from amplicon sequencing for all samples are available at SRA under accession code PRJNA673198. Processed data from amplicon sequencing as well as raw and processed data from the genetic screens are provided as supplementary files (Supplementary Files 2, 3, 4).

The following datasets were generated:

| Author(s) | Year | Dataset title | Dataset URL | Database and Identifier |
|---|---|---|---|---|
| Jost M, Jacobson AN, Fischbach MA, Weissman JS | 2020 | CRISPR genome editing of human dendritic cells (treatments of unedited dendritic cells) | https://www.ncbi.nlm.nih.gov/geo/query/acc.cgi?acc=GSE161401 | NCBI Gene Expression Omnibus, GSE161401 |
| Jost M, Jacobson AN, Fischbach MA, Weissman JS | 2020 | CRISPR genome editing of human dendritic cells (treatments of knockout dendritic cells) | https://www.ncbi.nlm.nih.gov/geo/query/acc.cgi?acc=GSE161466 | NCBI Gene Expression Omnibus, GSE161466 |
| Jost M, Jacobson AN, Hussmann JA, Fischbach MA, Weissman JS | 2020 | CRISPR genome editing of human dendritic cells | https://dataview.ncbi.nlm.nih.gov/object/PRJNA673198 | NCBI Gene Expression Omnibus, PRJNA673198 |

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
