## [Decision Letter]

**Acceptance summary:**

Jost and colleagues have developed a robust non-viral CRISPR/Cas9-based genome-editing methodology that enables to directly edit differentiated moDCs. This will open the possibility for functional genomics screening with DCs as well as targeted mechanistic studies that will enhance our biological understanding on the role of DCs for innate and adaptive immunity.

**Decision letter after peer review:**

Thank you for submitting your article "CRISPR-based functional genomics in human dendritic cells" for consideration by *eLife*. Your article has been reviewed by 2 peer reviewers, and the evaluation has been overseen by a Reviewing Editor and Satyajit Rath as the Senior Editor. The following individuals involved in review of your submission have agreed to reveal their identity: Klio Maratou (Reviewer #1); Carla V Rothlin (Reviewer #2).

Essential revisions:

1. As this manuscript comes under the category of "tools and resources article", it is important that the authors demonstrate that the editing is not changing the function of dendritic cells for the readers to have more confidence that the methodology is sound and will enable future studies on dendritic cells biology. Since the authors have already collected RNAseq data, they should capitalised on this dataset and conduct a more thorough analysis of this data. They need to compare the neutral control RPE65 KO vs. PBS samples to show that the editing does not significantly alter the transcriptomic profile of the dendritic cells. Can they see statistically significant differential expression changes between the two samples. Do these changes relate to pathways that are important for dendritic cell function? If so, what are these? Are any of the dendritic cell maturation or cell signalling pathways affected? Finally, it could be possible to perform immunophenotyping of the cells using fluorescent‐based flow cytometry with a panel of DC markers and other myeloid cell lineage markers to show that the process of editing does not de-differentiate/activate the cells or change their morphology.

2. To validate and apply this strategy, the authors conducted a genetic screen for factors relevant to the response to LPS from different species as well as drivers of interindividual variation. The study is by and large is carefully designed and includes appropriate controls such as for example editing of the chemokine receptor CXCR4 which is not involved in the response to LPS and therefore serves as a negative control. However, weaknesses are noted in the application of the strategy to identify receptors for LPS or drivers of interindividual variability. The authors claim that their genetic screen allows them to identify receptors for LPS derived from B. theta as well as drivers of interindividual variation. Unfortunately, at this stage, the data does not fully support the authors conclusions. The authors should validate the potential role of TLR2 in an independent setting, for example making use of a TLR2 reporter assay. This is of particular importance for concluding that TLR2 is a receptor for LPS derived from B. theta.

3. The response to B. theta varies in magnitude between donors even in non-edited cells. Is donor to donor variability too high to enable us to draw any biological conclusions when moDCs are used in screening? How many donors should be tested for biological phenotypic data to be meaningful?

As noted by the authors the maximal response to LPS in terms of TNF production is different between the donors. Also, it appears that EC50s are different. In order to establish drivers of interindividual variation, the authors should compare the effect of knocking out gene X in cells from individual A and B when stimulated with EC50 concentrations of LPS.

The authors emphasize differences in the effect of knocking out PTPN6 or IL-10 in the TNF secretion induced by LPS from B. theta. The authors go on to say (line 230) "Most prominently, knockouts of PTPN6 (SHP-1) and to a lesser extent IL10 increased TNF-α secretion in response to B. theta LPS in one donor ("donor i") but not the other ("donor h"), suggesting that these factors can constitutively suppress TNF-α secretion in a manner that differs among individuals (Figure 3d). Indeed, unedited moDCs from donor i secreted less TNF-α in response to both B. theta LPS and *E. coli* LPS than those from donor h (Figure 3e)." The fact that unedited moDCs from donor i secrete less TNF-α in response to LPS is not sufficient support that PTPN6 and IL-10 constitutively suppress TNF-α secretion. The authors should test if moDCs spontaneously secrete TNF if knocked out for PTPN6 or IL-10?

In various instances, the authors should consider discussing the limitations of the approach and temper their conclusions: In this regard, the authors should revise their biological conclusions and stress that this only a proof-of-concept study as the screen was conducted with limited number of donors. The authors should be very careful with the interpretation on whether the genes that are hits from the screen are true drivers of interindividual variability. The authors clearly demonstrate that moDC responses to LPSs vary across individuals. This point highlights the importance of assessing immune phenotypes in donor-derived cells, but also raises the necessity to use multiple donors for meaningful and reliable conclusions to be drawn from the data. The number of biological donor replicates (2-3) used in this study is sufficient to show proof of concept, but subsequent studies using moDCs will require much larger number of donors.

---

## [Author Response]

Essential revisions:1. As this manuscript comes under the category of "tools and resources article", it is important that the authors demonstrate that the editing is not changing the function of dendritic cells for the readers to have more confidence that the methodology is sound and will enable future studies on dendritic cells biology. Since the authors have already collected RNAseq data, they should capitalised on this dataset and conduct a more thorough analysis of this data. They need to compare the neutral control RPE65 KO vs. PBS samples to show that the editing does not significantly alter the transcriptomic profile of the dendritic cells. Can they see statistically significant differential expression changes between the two samples. Do these changes relate to pathways that are important for dendritic cell function? If so, what are these? Are any of the dendritic cell maturation or cell signalling pathways affected? Finally, it could be possible to perform immunophenotyping of the cells using fluorescent‐based flow cytometry with a panel of DC markers and other myeloid cell lineage markers to show that the process of editing does not de-differentiate/activate the cells or change their morphology.

We agree with the reviewers that it is important to demonstrate that our method does not perturb DC function and will enable studies of dendritic cell biology. We now include additional data to support this point.

First, we characterized edited and unedited DCs from two independent donors by flow cytometry, measuring levels of CD11c (a DC marker), HLA-DR and CD83 (two markers whose expression increases during DC maturation), TLR4 (expression of which decreases during DC maturation), and B2M (which we targeted for knockout, see below). We did not include additional markers because we already occupied 6 colors (including a viability stain) and we wanted to balance the ability to quantify marker levels against the breadth of the panel. As shown in Figure 1—figure supplement 5A, the staining patterns are qualitatively indistinguishable in knockout DCs and in unedited DCs (no electroporation control DCs or DCs electroporated with a non-targeting control sgRNA). We observed small differences between expression profiles of edited DCs and unedited DCs that were grown in flasks, pointing to changes in DC state as a consequence of culture conditions; such changes can be controlled by comparing knockout DCs to identically cultured unedited DCs or to DCs with knockout of a neutral gene, as we did throughout the manuscript. Overall, these results suggest that moDCs do not undergo de-differentiation or maturation as a consequence of the editing procedure.

We also stained for B2M, but we designed the panel for maximal resolution in the other channels, precluding accurate quantification of reductions in B2M levels upon knockout. We can place a conservative lower bound of >70% depletion for B2M. More broadly, our functional assays clearly demonstrate a reduction in protein levels in knockout cells.

The flow cytometry data are presented in Figure 1—figure supplement 5 in the revised manuscript and described in a new paragraph in the Results section. We also included details of the experimental strategy in the methods section.

To compare the transcriptional states of edited and unedited DCs, we compared RNA-seq data from mock-treated edited cells (donors p and q; data from Figure 2—figure supplement 2e) and from mock-treated unedited cells (donors r and s; data from Figure 2b). Clustering of the transcriptional profiles revealed that transcriptional states of edited DCs can be more similar to those of unedited DCs than those of unedited DCs from different donors to each other, suggesting that any changes induced by the editing procedure are smaller than any natural variation in DC state. In addition, the expression levels of a panel of DC markers are comparable between edited and unedited DCs. These analyses provide further evidence that the editing procedure does not substantially alter DC state. These data are included in the revised manuscript as Figure 2—figure supplement 3 and are described in the Results section.

We realize that this analysis may not be the one the reviewers had in mind. We could not perform the suggested direct comparison between unedited and RPE65 KO DCs because we did not include unedited DCs from the same donors in the RNA-seq experiment on knockout DCs. Indeed, throughout the manuscript we used DC populations with knockout of a neutral gene (generally RPE65) as the control populations to which we compared all other data. In our view this is the most robust way to ensure that the comparisons isolate the effect of the knockout.

Together with our data demonstrating that edited moDCs retain the ability to respond to diverse innate immune ligands (Figure 1f, Figure 1—figure supplement 3), the flow cytometry data and the additional analyses of the RNA-seq data provide strong support that our editing procedure does not perturb DC state and function.

2. To validate and apply this strategy, the authors conducted a genetic screen for factors relevant to the response to LPS from different species as well as drivers of interindividual variation. The study is by and large is carefully designed and includes appropriate controls such as for example editing of the chemokine receptor CXCR4 which is not involved in the response to LPS and therefore serves as a negative control. However, weaknesses are noted in the application of the strategy to identify receptors for LPS or drivers of interindividual variability. The authors claim that their genetic screen allows them to identify receptors for LPS derived from B. theta as well as drivers of interindividual variation. Unfortunately, at this stage, the data does not fully support the authors conclusions. The authors should validate the potential role of TLR2 in an independent setting, for example making use of a TLR2 reporter assay. This is of particular importance for concluding that TLR2 is a receptor for LPS derived from B. theta.

We agree with the reviewers that some of our claims needed further support to draw definitive conclusions. However, as the reviewers pointed out in the first and last main comments, the main innovation in our manuscript is the method to introduce knockouts in DCs. The genetic screen recapitulated known LPS signaling pathways and thus provided proof-of-principle for the utility and scalability of our method. The additional hypotheses emerging from the screen data, such as the potential roles of TLR2 in *B*. *theta* LPS recognition and of PTPN6 in mediating inter-individual variation, were intriguing but not essential to establish the utility. We therefore re-framed the manuscript to more clearly emphasize the methods development and proof-of-principle nature of the screen, while presenting the additional points as hypotheses for further exploration.

3. The response to B. theta varies in magnitude between donors even in non-edited cells. Is donor to donor variability too high to enable us to draw any biological conclusions when moDCs are used in screening? How many donors should be tested for biological phenotypic data to be meaningful?

First, although we observe that the magnitudes of the LPS response varies across DCs from different donors, many other core aspects of biology are consistent across donors. For example, the patterns of activity across different LPSs are consistent: *B*. *theta* WT LPS is a weaker agonist than *E. coli* LPS and *B*. *theta* 4PP LPS has further reduced activity in all donors. In addition, our genetic screen identifies many core factors that are important for LPS recognition across donors. Thus, many biological conclusions can be drawn from moDCs by interrogating cells from 2-4 different donors, as we did throughout the manuscript.

Second and perhaps more broadly, inter-individual variability is an interesting biological phenomenon in itself, but how it arises from environmental and genetic factors has long been difficult to study. In this context, the ability to conduct genetics in moDCs from different individuals will be essential to pick apart these contributions, for example through longitudinally sampling cells from the same donor to disentangle genetic and environmental contributions. We provide first glimpses of how we can begin understanding the drivers of such variation by comparing cells from 2-4 different donors. We agree that to definitively pinpoint drivers, studies in larger numbers of donors will be needed, as further discussed below. Importantly, we observed high knockout efficiency across all donors (~15 total), suggesting that the utility of our method is not limited by inter-individual variation and will be applicable to studies involving large numbers of donors.

We touch upon these points in the Discussion section of the revised manuscript.

As noted by the authors the maximal response to LPS in terms of TNF production is different between the donors. Also, it appears that EC50s are different. In order to establish drivers of interindividual variation, the authors should compare the effect of knocking out gene X in cells from individual A and B when stimulated with EC50 concentrations of LPS.

We used non-saturating concentrations of LPS for all experiments with knockouts. We do not think we would be able to learn more about the drivers of inter-individual variation by stimulating at exactly EC50.

The authors emphasize differences in the effect of knocking out PTPN6 or IL-10 in the TNF secretion induced by LPS from B. theta. The authors go on to say (line 230) "Most prominently, knockouts of PTPN6 (SHP-1) and to a lesser extent IL10 increased TNF-α secretion in response to B. theta LPS in one donor ("donor i") but not the other ("donor h"), suggesting that these factors can constitutively suppress TNF-α secretion in a manner that differs among individuals (Figure 3d). Indeed, unedited moDCs from donor i secreted less TNF-α in response to both B. theta LPS and *E. coli* LPS than those from donor h (Figure 3e)." The fact that unedited moDCs from donor i secrete less TNF-α in response to LPS is not sufficient support that PTPN6 and IL-10 constitutively suppress TNF-α secretion. The authors should test if moDCs spontaneously secrete TNF if knocked out for PTPN6 or IL-10?

We did not clearly state our model. Our hypothesis is that PTPN6 and IL-10 suppress TNF-α secretion upon stimulation of the cells, such that knockout of *PTPN6* or *IL10* increases TNF- α secretion upon stimulation. We re-phrased the corresponding section to clarify. More broadly, we now more clearly emphasize that this hypothesis requires further exploration.

In various instances, the authors should consider discussing the limitations of the approach and temper their conclusions: In this regard, the authors should revise their biological conclusions and stress that this only a proof-of-concept study as the screen was conducted with limited number of donors. The authors should be very careful with the interpretation on whether the genes that are hits from the screen are true drivers of interindividual variability. The authors clearly demonstrate that moDC responses to LPSs vary across individuals. This point highlights the importance of assessing immune phenotypes in donor-derived cells, but also raises the necessity to use multiple donors for meaningful and reliable conclusions to be drawn from the data. The number of biological donor replicates (2-3) used in this study is sufficient to show proof of concept, but subsequent studies using moDCs will require much larger number of donors.

We agree with the reviewers that a discussion of the strengths and limitations of the approach was missing from the manuscript and have included such a discussion in the revised manuscript at the beginning of the Discussion section.

We also agree that studies in larger numbers of donors will be needed to definitively establish the roles of putative drivers of inter-individual variation, which we also note in the revised Discussion. More broadly, we now emphasize that our study provides proof-of-principle, suggests exciting hypotheses to be explored further, and provides enabling tools for such explorations.